

# An integrated modeling framework for coevolution and feedback loops of nexus across economy, ecology and food systems based on the sustainable development of water resources

Yaogeng Tan[1,2], Zengchuan Dong[1], Sandra M. Guzman[2], Xinkui Wang[1], Wei Yan[3]

1. College of Hydrology and Water Resources, Hohai University, Nanjing 210098, China

2. Department of Agricultural and Biological Engineering, Indian River Research and Education Center, University of Florida, Fort Pierce, FL 34945, United States.

3. School of Geographic Sciences, Xinyang Normal University, Xinyang, 464000, China

Correspondence to Yaogeng Tan (170201010014@hhu.edu.cn)

**Abstract:** Sustainable development in water resources is becoming a hot topic in recent years. The world is facing the disequilibrium between the availability of resources and the increase in population with fast-growing economies and social development. This study proposes a new methodological framework of sustainable development of water resources based on the response linkages and feedback loops of economy-ecology-food (EEF) nexus. It provides a new way to identify the interconnection and coevolution process between these EEF. The multi-objective model and system

dynamic (SD) model were coupled to characterize the interconnections between processes and their dynamic responses to a set of scenarios. The combination of decomposition-coordination method (DC) and dynamic programming was used to find the optimal scenario based on each component of the EEF nexus. The Upper reach of Guijiang River Basin (UGRB) was presented as a case study. Results showed that the coupled multi-objective model and SD model presented in this study are able to characterize the interactions and feedback between EEF systems adequately. Most importantly,

the rapid growth rate of socio-economic indexes will drive the awareness of river ecology and showed a higher sensitivity under different decision preferences. The results provided in this study can provide baseline information for stakeholders and policymakers in the field of water management for a better understanding of the interactions across systems.

**Keywords:** Sustainable development; economy-ecology-food nexus; Optimal modeling; system dynamic model; coevolution; feedback loops

## 1. Introduction

As global warming caused by climate change and growing population, the world is facing the disequilibrium between natural resources sustainability and human wellbeing (Zhang et al., 2018; Luo and Zuo, 2019; Bei et al., 2009; Yang et al., 2019). In many regions, anthropogenic activities have led to an enormous demand for natural resources, which may have a negative influence on future population development. Recently, there has been an increased interest

in "sustainable development" because of its ambiguity and applicability in both local and global environments (Biggs et al., 2015; Duan et al., 2019). The new targets of sustainable development aim to achieve sustainable uses of water resources, energy, and sustainable agricultural practices and promote inclusive economic development (United Nations, 2014). As the irreplaceable foundation of social development and environmental protection, water is one of the most critical natural resources and plays a vital role in socio-economic development and human production (Walter et al.,

2012; Yang et al., 2018).

The concept of sustainable development was first presented in the World Conference on Environment and Development (WCED) (Brundtland, 1987). The goal of this conference was to discuss how to achieve a systematic development of the economy and environment, based on environmental resources protection. According to the assumption of sustainable development, the sustainable use of water resources needs population requirements, but not

at the expense of finite sources of water (Lant, 2004). The core content of sustainable development of water resources is





to serve both socio-economic and ecological components (Flint, 2004; Konar et al., 2016). Similarly, sustainable use of water resources should be a combination of economy and environment (Rogers et al., 2002). Previous studies have evaluated options to maintain sustainable development. According to Vorosmarty et al. (2010), the purpose of developing infrastructure is to ensure the safety of water use, that is, to meet human water needs as much as possible. However, it

damages the effect of water resources on ecological environments and does not conform to the basic concept of sustainable development. Hoekstra et al., (2014) also explained that water usage based on socio-economics should be under a certain upper limit in water allocation process to meet the requirements of ecological environments and further achieve the sustainable development of water resources. However, the components of both socioeconomics and environmental stewardship are complex, and actions should be taken to deepen the insights of sustainable development.

For example, socioeconomics also has underlying interconnections with industrial and agricultural practices. These practices require water resources to make profits. Agriculture is the largest consumer of freshwater and contributes to food and crop production (Li et al., 2019). Likewise, environmental stewardship is a complex system that maintains essential function and biodiversity in freshwater sources, vegetation, and ecological processes. Thus, the utilization of water in complex systems has an impact on the corresponding interconnected processes. From this view, these "nexus"

or interactions between processes and systems should be considered to better understand the term of "sustainable development".

The "nexus thinking" was first conceived by the World Economic Forum (2011) to promote and discuss the indivisible relationships between the use of resources and providing the universal rights of water, energy and food (Hoff, 2011; Biggs et al., 2015). Furthermore, the framework of water-energy-food (WEF) nexus is propelled, which has drawn

an extensive attention (Allam and Eltahir, 2019; Sarkodie and Owusu., 2020). WEF refers to the complex interlinkages among these three items, in pursuit of sustainable development (Mabhaudhi et al., 2019). "Nexus thinking" is essentially the coupling of interconnecting systems, and coevolution is its crucial process. It is a "dynamic" model where the optimization decisions can evolve accommodatively as the reactions to the changes of other subsystems (Feng et al., 2019; Wang et al., 2016; Di Baldassarre et al., 2015), instead of "static" model on which the impact of other systems are

simulated by scenario-based analysis (Rheinheimer et al., 2016; Perrone and Hornberger., 2016). Nexus thinking presents a new way of approaching WEF crises, and these three items are essential for human welfare and social, economic, and environmental aspects of sustainability (Foden et al., 2019; Ghani et al., 2019). It constitutes a framework for analyzing the interaction between the three of them (Liu et al., 2017; Niva et al., 2020). The concept of WEF nexus has been reported in different resources management and global regions in the last decades (Smajgl et al., 2016; Kantor et al.,

2017; Franz et al., 2018; Sharifi et al., 2019). Apart from WEF nexus, other aspects of "nexus thinking" was also conceived by researchers and also contribute to sustainable development, such as land use–water–energy linkages, of which the food is core component (Howells et al., 2013; Ringler et al., 2013). Hellegers et al., (2008) outlined the concept of sustainability by combing water, energy, food, and environment, which can be regarded as the nexus thinking of water-energy-food-environment. This literature presented the urgency and necessity to assess the interaction of four

components to minimize the negative effects and enhance the synergy benefits. Ren et al. (2018) achieved integrated and sustainable development by modeling land-water-food nexus in North China. Shahzad et al., (2017) stressed that water and energy are closely interlinked, and utilization of both resources will lead to increase $CO_2$ emissions and environmental risks, and fulfill future sustainability by energy-water-environment nexus. Feng et al. (2016; 2019) outlined the framework of water, power and environment systems (WPE nexus), disclosed their coevolution and response

linkages of these three items, and gave a strong reference for policymakers.

According to these researches, the concept of "sustainable development" in recent years refers to multiple disciplines and aspects, including hydrology, agriculture, anthropology, geography, geology, economy, ecology, energy, etc., and their free combinations. It refers to many different goals and their mutual connections should be essentially considered in policy formulation and implementation using a systems approach (Pahl-Wostl, 2019). Therefore,





computational modeling is often a powerful method to quantify the mutual effect of either WEF or other nexus in the process of sustainable decision making. Some studies have proposed a comprehensive mathematical model for managing different resources and analyzing the inseparable interlinkages across WEF nexus over the years. The theory of Complex Adaptive System (CAS) is developed based on the systematic theory to tackle such problem as "nexus" is substantially a complex system (Holland, 1992). CAS consists of several adaptive agents that have their own goals and can change

their behaviors to attain co-exist and best status by adaptive self-learning and "accumulate experiences" from external changing environment. The Agent-based modeling (ABM) is also an effective approach for complex system model as each discrete and individual agent has different objectives and behaviors under different social backgrounds (Zhang et al., 2018; Macal and North, 2010). Multi-objective programming model is a promising tool to assess methodologies to achieve the goal of sustainable development as well as their subsystem nexus because adaptive process is substantially

optimization among multiple objectives. For example, Li et al., (2019) developed an optimal model called Agricultural Water-Energy-Food Sustainable Management (AWEFSM) to address the tradeoffs between water & land resources and energy to generate the environmentally friendly strategies and policies. Feng et al., (2016; 2019) developed a set of models considering water supply, environmental and power generation to produce a parallel development. Khan et al., (2017) addressed the impact of water management decision on water-energy-food-environment nexus on the basin scale

by coupling SWAT and water system model through ABM framework. Considering systems nexus is mostly nonlinear, advanced optimal programming is also used as part of the multi-objective programming. Some examples of advanced optimal programming include dynamic programming (Li et al., 2015), genetic algorithms (Chang and Chang, 2009; Bai et al.,2015), and decomposition-coordination programming (Jia et al., 2015; Tan et al., 2019). Furthermore, decision makers often need to find the optimal framework to achieve sustainable development by evaluating how each system

and subsystem changes under external conditions, taking into account uncertainty in future scenarios (Wang et al., 2019). According to Biggs et al., (2015) and Pahl-Wostl, (2019), the key procedure of achieving sustainable development in recent years is to develop nexus thinking, and the nexus components can include several of the following processes including water, energy, food, land, environment, ecology, economy, agriculture, etc. However, previous studies provide a limited discussion on the nexus between economy, ecology, food from a systematic perspective taking into account

coevolution mechanisms to further advance into the goal of sustainable development. In this view, not only the multiple processes and their mutual connections should be fully considered, but also coevolution and feedback should be investigated to further achieve sustainable development of water resources by assessing the tradeoffs across objectives.

       In this study, we present a theoretical framework of water resources sustainable development based on economy-ecology-food (EEF) nexus, and implement model optimization to assess the tradeoffs among different scenarios to

achieve the goal of each part of the EEF nexus. Then, the system dynamic (SD) model is established to evaluate the endogenous dynamics and feedbacks across each objective of EEF nexus, and further evaluate the degree of sustainable development by setting up an evaluation index system based on each part of EEF nexus. The proposed integrated model is adopted to a case study in the Upper reaches of Guijiang River Basin, China. Results demonstrate how the integrated model and theoretical framework can provide useful information for the stakeholders to achieve sustainable development

for water resources and provide insights for water resources management among different goals and its effect on human lives.

## 2. Methodology

### 2.1 Outline of EEF nexus and its main modules

       The economic, ecological and food (EEF) influences on water resources sustainability were evaluated through the

development of an EEF nexus model. The framework of sustainable development is divided into three modules: economy, ecology and food, and their mutual relationship are shown in **Fig.1**. The three modules are intercorrelated, and the EEF





nexus motivates policymakers to analyze tradeoffs between different processes or objectives and further adjusts the optimal development mode. Arrows represent the outputs of each module, which indicates the level of impact in sustainable water development. Food and economic module attain their own target by development and controlling water resources that supply water to ensure social activities and food safety. At once it is also indispensable for ecological module that maintains ecological functions, which restricts the appliance to both economic and food module. Therefore, optimal allocation is processed to take into account each module to attain sustainable development of water resources (See 2.2). Each module contains three submodules or subsystems that are interconnected based on the module goal.


**Fig.1** Theoretical framework of EEF nexus






### 2.1.1 Economic module

This module is used to determine the socio-economic and water interactions, including water withdrawal, usage, consumption and drainage (Luo and Zuo, 2019). It should be noted that social element is included in this module because the relationship between society and economy is usually inseparable. From the water supply perspective, it also supplies water for social (household) use. The household and industrial water demand are presented as follows:

$$WD_{hou} = \frac{q_{hou} \times N \times d}{1000} \qquad (1a)$$

$$WD_{indus} = I_{GDP} \times q_{indus} \qquad (1b)$$

$$\frac{dN}{dt} = rN \qquad (1c)$$

where $WD_{hou}$ and $WD_{indus}$ are the annual household and industrial (including secondary and tertiary) water demand (m³), N is population size, d is the days of a certain year and r is the natural growth rate, $I_{GDP}$ is the industrial added value (10⁴ Yuan), $q_{hou}$ and $q_{indus}$ are the domestic and industrial water usage quota, which means daily water consumption per person (L/person/day) and water consumption of the industrial added value per 10⁴ Yuan (m³/10⁴ Yuan), respectively. The population equation presented in Eq.(1c) is a simple linear differential model called Malthusian growth model (Jørgensen and Bendoricchio, 2001; Feng et al., 2016), and GDP size is also suitable for this model. The population size changes are based on the assumption of the socio-hydrological system (See below). The objective of this module is to minimize the water shortage of both human's live and industrial production, and is the necessary condition to make the maximum carrying capacity (description shown in Section 2.3.1) of population and GDP. The main variables index list is shown in Appendix A, including the model variables presented afterwards.

Many researchers conceived the coevolution process of socio-hydrological system, including "Taiji-Tire model" (Liu et al., 2014), "community sensitivity model" (Elshafei et al., 2014) and "pendulum swing model" (Van et al., 2014; Kandasamy et al., 2014). The social development is at the expense of sacrificing the environment, and the "pendulum model" is therefore addressed based on different development stages over the past years and adapted in Australia. Kandasamy et al., (2014) stressed that the term "pendulum swing" refers to the shift in the balance of water utilization between economic development and environmental protection. The agricultural-based society is at the beginning of the evolution, and the environmental problems have not emerged in this stage. As water resources benefit to both agricultural and socio-economic development with massive government policy support and investment, the whole society's demand for resources has intensified due to the sharp growth of population as a result of increased irrigation area and agricultural productivity, and furthermore, the environment will be significantly damaged, which can be regarded as the pendulum "swings" towards the economic development. When environmental awareness is on the rise, the government will invest more in ecology, resulting in a declining population. In this case, more water is used to protect the environment, reflecting that the pendulum has "swung" back to the environment.

### 2.1.2 Ecological module

#### (1) Ecological water demand for vegetation

Ecological water demand of vegetation is used to maintain the physiological function of canopies, including photosynthesis, respiration and evapotranspiration. The method of evaluating the amount of vegetation ecological demand is based on their evapotranspiration that can be treated as the water gap (Shi et al., 2016; Saxton et al., 1986):

$$WD_{veg} = K_s \cdot K_c \cdot ET_0 - P_e \qquad (2a)$$



$$ET_0 = \frac{0.408\Delta\left(H_{net} - G\right) + \gamma\frac{900}{T+273}u_2\left(e_0 - e_z\right)}{\Delta + \gamma\left(1 + 0.34u_2\right)} \quad (2b)$$

$$K_s = \frac{\ln\left[100\times\dfrac{S - S_w}{S_c - S_w} + 1\right]}{\ln 101} \quad (2c)$$

where $WD_{veg}$ is the vegetation water demand. $ET_0$ is potential evapotranspiration based on the Penman-Monteith equation, and the particular variables can be seen in Neitsch et al., (2011). $K_s$ and $K_c$ are soil moisture and canopy coefficients, respectively, which denotes the ratio of maximum water demand and potential evapotranspiration. S, $S_c$ and $S_w$ are the coefficient of actual, wilting and critical soil moisture, respectively. $P_e$ is effective precipitation and is calculated based on the following (Döll and Siebert., 2002):

$$P_e = \begin{cases} P\times\left(1 - \dfrac{0.2P}{125}\right), & \text{if } P \le 250 \\ 0.1P + 125, & \text{if } P > 250 \end{cases} \quad (2d)$$

where P is actual precipitation.

(2) River ecological demand

River ecological demand is the instream water demand that is used to maintain river health and function. Its health degree can be reflected by the annual proportional flow deviation (APFD) that is used to assess the diversity of fish
species (Gehrke et al., 1995). However, it is computationally unstable when the monthly streamflow is near zero (Yin et al., 2010). In this study, we use the amended indicator, AAPFD, to assess the river ecological demand (Ladson and White, 1999):

$$AAPFD = \frac{1}{n}\sum_{j=1}^{n}\sqrt{\sum_{m}^{12}\left(\frac{Q_{mj} - QN_{mj}}{\overline{QN_j}}\right)^2} \quad (3)$$

where Q and QN are the actual and natural streamflow. The subscript n, m and j are the total year number, mth month,
and jth year. According to Ladson and White, (1999), the smaller deviation suggests the better river ecology, which is reflected by smaller AAPFD, and the value of AAPFD ranges from zero to five. When the value is larger than five, the river ecosystem will be seriously damaged (Yin et al., 2010; Tan et al., 2019). Therefore, the goal of evaluating the river ecological demand is to find a suitable Q to make AAPFD minimum.

(3) Sewage water
The water cycle from the socio-economic module in **Fig.1** includes the water discharge as one of the outputs. This water can be reused for water supply in other processes especially for socioeconomics, and make more efficient water treatment and use of recycled water. The total amount of recycled water resources is expressed by:

$$W_{reuse} = \left(\alpha_1\beta_1 WD_{hou} + \alpha_2\beta_2 WD_{indus}\right)\cdot\zeta \quad (4)$$

where α and β are sewage water drainage coefficient and sewage water treatment rate, respectively. The subscript 1 and
2 is household and industrial water usage. ζ is the utilization rate of recycled water.

*2.1.3 Food module*

The food module is mostly related to agricultural water usage, including crop water requirements based on phenological stages and farm management including livestock production. For crops, water usage is related to crop yield.





The main water supply is provided by irrigation. We use the crop coefficient method to estimate crop water demand
based on the Food and Agricultural Organization report No. 56 (FAO-56) (Allen et al., 1998). For each crop, its growth
process can be separated into several stages that have the different potential crop water demands (Allen et al., 1998;
Smilovic et al., 2016):

$$W_p = \int_{t_0}^{t_n} K_c(t) \cdot ET_0 dt \tag{5a}$$

$$W_a = W_p - P_e \tag{5b}$$

where $W_P$ is potential crop water demand, and can also be called reference crop demand of crop i, $K_c(t)$ is the crop
coefficient of stage t for a specific crop, $t_0$ and $t_n$ is the first and last stage of the growth process of a specific crop. $W_a$ is
the irrigation water demand. The maximum crop yield is based on the hypothesis that the crop water supply (including
precipitation) can meet $W_p$ (Allen et al., 1998). According to FAO-56, crop growth is usually divided into four
phenological stages: initial, development, middle and end, and corresponds to three different crop coefficients: $K_{c,ini}$,
$K_{c,mid}$ and $K_{c,end}$. For details, see Allen et al., (1998). For each crop, the crop yield is presented as follow (Smilovic et al.,
2016):

$$\frac{Y_s}{Y_p} = \prod_{t=t_0}^{t_n} \frac{Y_{s,t}}{Y_{p,t}} = \prod_{t=t_0}^{t_n} \left[ 1 - K_{y,t} \left( 1 - \frac{W_{s,t} + P_{e,t}}{W_{p,t}} \right) \right] \tag{6}$$

where $W_{s,t}$ is the actual irrigation water supply for crop i at time t, $Y_s$ and $Y_p$ is the crop yield under actual and ideal
condition (both irrigation water supply $W_s$ and precipitation $P_e$ can meet the crop water demand $W_p$), $K_{y,t}$ is yield response
factor of the crop i at time t. Due to the limitation of local water resources conditions, crop water supply is usually equal
to or less than crop water demand. That is, $(W_s + P_e) \leqslant W_p$, and crop water supply is greatly related to crop yield. The
value of $Y_s/Y_p$ is also equal to or less than one, and it takes the "=" sign when the crop yield attains the maximum. In
this case, the water supply also attains the maximum.

For meat production, it is reflected by the production of livestock (pork and beef) and poultry (chicken, duck and
goose). The calculation of water usage of livestock is the same as Eq.(1a) and here N and q are the total livestock
population and its corresponding water use. The production of livestock and poultry can be solved by linear regressive
calculation based on local statistical yearbooks and water resources bulletin over the historical years (Li et al., 2019):

$$Y_L = a_L W_L + b_L \tag{7}$$

where $Y_L$ is the production of a certain livestock ($10^4$t) and $W_L$ is the actual water use of a certain livestock ($10^4$m$^3$), $a_L$
and $b_L$ are primary coefficient and constant term of the stock-water production function.

*2.2 Distributed optimal model of sustainable development based on EEF nexus*

*2.2.1 Model conceptualization*

The framework of sustainable development theory presented in **Fig.1** is of great significance by applying it in a
specific region or watershed. For example, in a water system inside a watershed or a region, there are multiple water
supply projects within which water users are interconnected. This system in a watershed is called a "large water resources
system" (**Fig.2**a). It is subdivided into multiple sub-watershed or subregions that are called "subsystems" (**Fig.2**b). In
this case, reservoirs can provide not only socio-economic developments but also environmental impacts. They are
constructed across the rivers to supply water for the whole region or watershed but are also most likely to cause negative
impacts on the natural streamflow of rivers, which will deteriorate the instream ecological environment (Yin et al., 2010;
2011; Yu et al., 2017). Therefore, reservoirs should be considered individually to target the river ecology concerns.


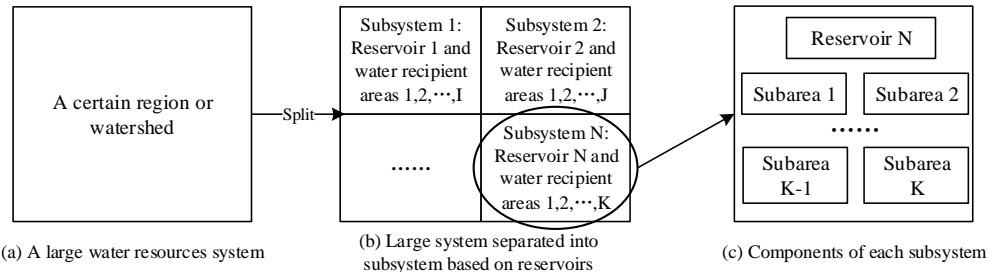

(a) A large water resources system  (b) Large system separated into subsystem based on reservoirs  (c) Components of each subsystem

**Fig.2** Water resources system and its decomposition

To fully consider the river ecological health, the whole system is separated into subsystems that contain one individual reservoir and its several corresponding water recipient areas (**Fig.2**b) as there is usually more than one

reservoir in a certain region. We call these subsystems as "reservoir supply subsystem". A subsystem can be further separated into the smallest unit: a reservoir and each water recipient region (or called "subarea") (**Fig.2**c). In this view, the total system of sustainable development of water resources in a certain region (watershed) can be divided into several subsystems or subareas that consist of a three-level hierarchical structure. According to the theoretical framework of sustainable development of EEF presented in Section 2.1, each module has their own goals, and they can be distributed

to each subarea (with the objective of food, socio-economy and vegetation) and reservoir (river ecology) (**Fig.3**). Therefore, we can coordinate these objectives to achieve sustainable development by setting up multi-objective optimal model.



**Theoretical Framework**

Goal: Vegetation & river ecology

Ecology

Sustainable development of water resources

Economy

Food

Economy-Ecology nexus

Food-Ecology nexus

Goal: maximum carrying capacity & minimum water shortage

Economy-Food nexus

Goal: Crop Yield & Meat production

**Application: System generalization**

Water resources system of a certain region

Reservoir supply subsystem 1

Reservoir supply subsystem M

Reservoir 1 → Ecological module: River objective

Subarea 1 → Ecological module: Vegetation objective

Subarea 1 → Economy module: GDP & household goal

Subarea 1 → Food module: Crop yield & meat

Subarea $N_I$ → Ecological module: vegetation objective

Subarea $N_I$ → Economy module: GDP & household goal

Subarea $N_I$ → Food module: Crop yield & meat

Reservoir 1 → Ecological module: River objective

Subarea 1 → Ecological module: Vegetation objective

Subarea 1 → Economy module: GDP & household goal

Subarea 1 → Food module: Crop yield & meat

Subarea $N_k$ → Ecological module: vegetation objective

Subarea $N_k$ → Economy module: GDP & household goal

Subarea $N_k$ → Food module: Crop yield & meat

Upper Level: Large water resources system

Middle-Level: Reservoir supply subsystem

Lower-Level: individual reservoir and subareas

**Fig.3** Application of sustainable development theory on watershed or region

*2.2.2 Objective function*

(1) Economic module

The objective function is presented based on each component of the EEF nexus. The goal of the economic module is aiming at increasing revenue of secondary and tertiary industries, as well as maintaining human wellbeing. It can be reflected by the minimum household and industrial shortage and is expressed by the following normalized nonlinear

equation:





$$F_{ecnmy} = \frac{1}{T}\min\sum_{k=1}^{K}\sum_{t=1}^{T}\left(\frac{WD_{ecnmy,kt} - WS_{ecnmy,kt}}{WD_{kt}}\right)^2 \tag{8}$$

where $F_{ecnmy}$ is the objective function of economic module. WS and WD is total water demand (including household and industry) and supply (including reservoir and other water projects) of this module. T is the total number of the time step. Subscript k and t are the number of subarea and time step, respectively.

(2) Ecological module

Economic development should not be excessive because it may be at the expense of the damaging ecological environment, which is inconsistent with the concept of sustainable development. Similar to maintaining human wellbeing and increasing the revenue of industries, water resources support is also indispensable for maintaining the physiological function of vegetation and river health. The objective of the ecological module is reflected by maintaining both aspects.

It should be noted that for recycled water usage, it is acted as the part of the water supply (WS) for economic module. The expression of objective function for ecological module is as follows:

$$F_{eclgy} = \frac{F_{veg} + F_{riv}}{2} \tag{9a}$$

where $F_{eclgy}$ is the objective function of ecological module, and

$$F_{veg} = \frac{1}{T}\min\sum_{k=1}^{K}\sum_{t=1}^{T}\left(\frac{WD_{veg,kt} - WS_{veg,kt}}{WD_{veg,kt}}\right)^2 \tag{9b}$$


$$F_{riv} = \min\frac{AAPFD}{5} \tag{9c}$$

where the subscript "veg" and "riv" represents vegetation and river ecology. According to Ladson and White (1999), the value of AAPFD ranges from zero to five. Here, we divided it by 5 to normalize AAPFD and make it range from zero to one. Meanwhile, $F_{eclgy}$ is also normalized by getting the average of $F_{veg}$ and $F_{riv}$.

     (3) Food module

The goal of the food module is to maximize food production and is the indispensable condition of increase primary industry products and maintain human wellbeing. The mathematical expression is presented as follow:

$$F_{food} = F_{crop} + F_{livestock} \tag{10a}$$

where

$$F_{crop} = \max\sum_{n=1}^{N}\left(\frac{Y_a}{Y_p}\right)_n \tag{10b}$$


$$F_{livestock} = \max\sum_{l=1}^{L}Y_L \tag{10c}$$

where N and L are the total number of crops and livestock, respectively.

*2.2.3 Tradeoffs between objectives*

According to Eq.(6), crop production is directly related to irrigation water (FAO, 2012; Liu et al., 2002; Lyu et al., 2020), and the production of livestock is also in proportion to its water usage (see Eq.(7)). Therefore, the maximum

supply of crop and livestock water demand is the most critical condition to get the maximum crop yield or meat production. Therefore, the normalized objective of food module can be rewritten as:





$$F_{food} = \frac{1}{T} \min \sum_{k=1}^{K} \sum_{t=1}^{T} \left( \frac{WD_{food,kt} - WS_{food,kt}}{WD_{food,kt}} \right)^2 \tag{11}$$

where $WS_{food}$ and $WD_{food}$ are the irrigation or livestock water supply & demand. Similarly, the maximum satisfaction of industrial and household water demand can get the maximum profit and revenue as well as human wellbeing, which is the same as the minimum water shortage. The same also applies to vegetation water.

As can be seen in objective functions, three benefits are set minimum (Eqs.(8)(9a)(11)), which may contribute to the conflict between objectives, especially ecology and economy. The tradeoffs across EEF nexus can be reflected by Pareto frontier that can describe a set of non-dominated optimal solutions that any one of these three objectives are unable to be improved unless sacrificing other objectives (Reddy and Kumar, 2007; Feng et al., 2019; Beh et al., 2015; Burke and Kendall., 2014). To overcome this problem, three major methods are used to generate Pareto frontiers: evolutionary algorithms (Reddy and Kumar., 2007), ε-constraint method (Haimes et al., 1971) and weighted-sum method (Marler et al., 2009; Burke and Kendall., 2014). Considering the continuous curve of Pareto frontiers is tough to be obtained by ε-constraint method, and coverage of local optima is often prone to obtained with evolutionary algorithms (Liu et al., 2011; Burke and Kendall, 2014), the weighted-sum method is adopted in this study.

We can reclassify all the water users from each of the three modules into two categories: Instream and off-stream water users (Hong et al., 2016). River ecological water demand (see Section 2.1.2) can be regarded as an instream water user and all others can be regarded as off-stream water users. Therefore, according to the objective function expressed by Eqs.(8),(9) and (11), the weighted objective function can be rewritten by:

$$\min F = F_{ecnmy} + F_{eclgy} + F_{food} = \alpha \left( F_{ecnmy} + F_{veg} + F_{food} \right) + \theta F_{riv}$$
$$= \sum_{j=1}^{J} \sum_{k=1}^{K} \sum_{t=1}^{T} \alpha_j \left( \frac{WD_{jkt} - WS_{jkt}}{WD_{jkt}} \right)^2 + \theta \frac{AAPFD}{5} \tag{12}$$

where ($F_{ecmny}+F_{veg}+F_{food}$) is off-stream water users, and $F_{riv}$ is the instream water users. The subscript $j$ is the index of the off-stream water users, respectively. $j=1,2,3$ represents socio-economic, food and vegetation water usage, which corresponds to the subscript "ecnmy", "eclgy" and "food". $\alpha$ and $\theta$ are weight factors and $\sum_{j=1}^{J} \alpha_j + \theta = 1$. Previous literature demonstrated the optimal solution shaped like Eq.(12) is Pareto-optimal because of the positive weights and concave objectives, and the non-dominated sorting process is used to find the optimal solution of Eq.(12) because the characteristic of either concave or convex is difficult to be proven (Marler and Arora., 2009; Feng et al., 2019; Goicoechea et al., 1982; Zadeh, 1963). For each given combination set of $\alpha$ and $\theta$, the optimal solution can be attained by decomposition and coordination (DC) principle and dynamic programming (DP) (see Section 2.2.5).

The tradeoff across objectives is reflected in the values of multiple sets of weighting factors $\left( \alpha_1, \alpha_2, \alpha_3, \theta \right)^T$, revealing different decision makers' preferences. Considering that the contradictions also occur in off-stream water users, the balanced priority should be addressed to give consideration for each off-stream water users (Casadei et al., 2016), that is, $\alpha_1 = \alpha_2 = \alpha_3$. Therefore, the tradeoff and decision preference between instream and off-stream is reflected by the different value of $\theta$ ($0 \leqslant \theta \leqslant 1$). The larger value of $\theta$ represents more concerns about river ecology. In this study, the parameter $\theta$ is initially set as 0.5 to give an equal consideration of both instream and off-stream water usage, and different levels of $\theta$ can affect the performance of EEF nexus and are used to assess the sensibility and uncertainty of the model (see Section 3.3.5).



*2.2.4 Constraints*

The model constraints include the connection of subsystems, the water balance equation, and the upper and lower limits. The details are found in Supplementary material S1 in Appendix B.

*2.2.5 model solution*

The EEF model of water resources sustainability is a compound system that is classified into multiple hieratical structures (**Fig.3**). Therefore, the model solution of this structure should be solved by systematical analysis techniques, such as Dantzig-wolfe decomposition technique (Deeb and Shahidehpour, 1990), Generalized Bender Decomposition (Rabiee and Parniani., 2013), aggregation-decomposition (AD) (Tan et al., 2017) and decomposition-coordination (DC) (Li et al., 2015; Jia et al., 2015). Considering DC method can reduce the system dimension to save computing time, and

optimization order among each subsystem is arbitrary, this study uses DC method to solve this sophisticated model. The total procedure of both DC and DP is provided in Supplementary materials S2 in Appendix B.

*2.3 Coevolution and responses of EEF nexus based on system dynamic (SD) model*

*2.3.1 Coevolution mechanism for each component of EEF nexus*

Water resources provide the resources support for agriculture (food module), industry (economy module) and

environment (ecology module). These components can, respectively, provide the crops and meat to ensure food security, making profit, and make human and nature co-exist harmoniously. The mutual relations among the three components of an EEF nexus determines the coevolution process (Feng et al., 2016). According to the framework of EEF nexus presented in section 2.1, the coevolution and responses of EEF nexus is shown in **Fig.4**.

As shown in **Fig.4**, the socioeconomic development, along with the population and GDP size, will undoubtedly

increase (Biggs et al., 2015; Duan et al., 2019), which will be reflected in an increase in water demand (I). However, the ecosystem will be damaged due to the volume of water that is going to supply those increased population needs (II). Therefore, the optimization model presented in this study can provide information to coordinate the nexus between systems, provide a water allocation scheme based on each module's water requirements, and maintain the ecological health of rivers and freshwater sources (III). The population and GDP growth rate are unable to increase infinitely

because regional water resources are usually unable to carry a continuously exponential growing population size and GDP. We call this term as "carrying capacity" that is used to describe the rate of socioeconomical development under certain water resources conditions (Yang et al., 2019; Wu et al., 2018). It is determined by the amount of actual water supply and allocation in a certain year. The carrying capacity can reflect the development status of a region and can inversely affect the predicted socioeconomic indexes (IV). It can give references for policymakers for urban

comprehensive planning and can influence the process of coevolution and feedback of EEF nexus (V). In this study, we use the concept of "overload index" to illustrate the relationship between carrying capacity and predicted economic index (mainly for population and GDP) and is expressed as follow:

$$OI = \frac{PI}{CI} \tag{13}$$

where OI, PI, CI is the overload index, predicted economic indicator and carrying economic indicator (i.e. carrying

capacity). The overload index can be classified into five levels based on the value of OI and shown in **Table 1**. This feedback loop indicated that the rapid growth of the economy would deteriorate ecological health because the limited water resources in a certain area cannot afford the increasing socio-economy. Additionally, ecological health is an indispensable element of sustainable development. It will further decrease the carrying capacity, and the socio-economy will, therefore, be negatively influenced, stimulating the policymakers to readjust the scale of socio-economy.

**Table 1**   Classification of overload index level

| Value of OI | Overload index level |
| --- | --- |





| ≤0.7 | Well-loaded |
| 0.7~1.0 | Rational-loaded |
| 1.0~1.3 | Minor overload |
| 1.3~1.5 | Moderate overload |
| >1.5 | Serious overload |

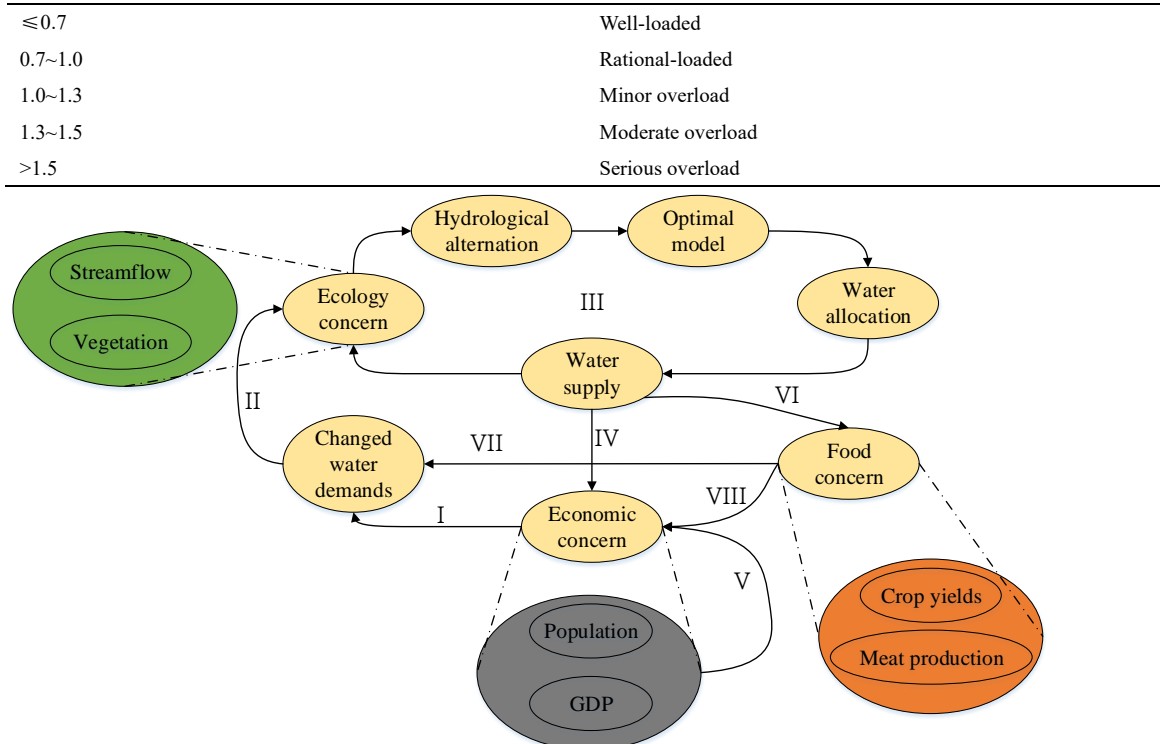

**Fig.4** Coevolution and feedback of EEF nexus

Another feedback of the EEF nexus is reflected by the ecology-food nexus. Agricultural water is the largest water consumer and is deeply affected by rainfall and potential evapotranspiration. According to Allen et al., (1998), more evapotranspiration will cause more agricultural water demand (VII), and water supply pressure from water projects will increase subsequently. However, if the rainfall increases, there will be less water supply pressure. Otherwise, the increased water supply from reservoirs will alternate the natural flow, which will deteriorate the river's ecological health and drive the optimal model to adjust the water allocation scheme (VII-II-III). Afterward, the agricultural water supply will affect food production (VI), which is similar to the effect that economy-ecology nexus reflects. However, the socio-economical changes would also indirectly affect the food system and more than just rainfall and evapotranspiration, i.e., the changes of economic concern will also drive the optimal model described in this study and further influences the food production (I-II-III-VI). Besides crop production, stock farming is another source of food for meat production and is also affected by this economy-ecology linkage. It should be noted that as food system is the indispensable component for human lives, the food production will directly affect the changes of carrying population (VIII) and subsequently affects the feedback loop of economy-ecology (I-II-III), further starting the new loop of whole EEF nexus.

From **Fig.4**, we can see that each component EEF nexus interacts mutually and are reciprocal causation. They are interconnected by the changes in water supply and demand system. To reflect the complicated and detailed relationships and feedbacks based on **Fig.4**, system dynamic model (SD) (Forrester and Warfield., 1971) is presented in this study. It is a well-established system simulation method for visualizing, understanding and analyzing complicated dynamic feedback systems that exhibit nonlinear, multi-feedback and time-varying properties (Yang et al., 2019). It can embody the framework of the detailed EEF nexus modules and can be seen as the detail and extension of the general framework of sustainable development (**Fig.1**). The detailed mutual relationships of each variable are shown in Supplementary materials S3 in Appendix B.



### 2.3.2 Sustainable development degree (SDD) assessment

The EEF nexus is a complex system with all ecological, economic and food systems, or modules as we called in this study, affecting water resources. A proper EEF balance provides resource support to achieve sustainable development. Therefore, the three modules should be considered to evaluate the sustainable development degree. We selected the indicators listed in **Table 2** based on the three modules and are used to evaluate the impact of sustainable development.

**Table 2** Sustainable development evaluation index system of three modules

| Module | Indicators | Source or calculating method | Property |
|---|---|---|---|
| Economy | Overload index of population | Eq.(13) | - |
| | Overload index of GDP | Eq.(13) | - |
| | Per capita GDP (RMB/people) | Carried GDP/Carried population | + |
| | Water consumption per 10000RMB of GDP ($m^3/10^4$RMB) | Total water supply/GDP | - |
| Food (Agriculture) | Meet production (t) | Eq.(7) | + |
| | Crop yield (t) | Eq.(6) | + |
| Ecology | Effective irrigation area for vegetation ($km^2$) | Vegetation water supply/Vegetation quota of crops | + |
| | AAPFD | Eq.(3) | - |
| | Sewage treatment rate | Eq.(4) | + |
| | Recycled water utilization rate | Eq.(4) | + |

The property (+, -) of indicators denotes positive and negative indicators, respectively. The positive (negative) indicators mean they have positive (negative) impacts on the corresponding module and were termed as a development (constraint) index (Yang et al., 2019). Considering the ranges of indicators listed in **Table 2** are different, they should be normalized before evaluation. The positive and negative indicators normalization is shown by Eq.(14a) and (14b).

$$y_{ij} = \frac{x_{ij} - \min\limits_{i=1}^{m} x_{ij}}{\max\limits_{i=1}^{m} x_{ij} - \min\limits_{i=1}^{m} x_{ij}} \tag{14a}$$

$$y_{ij} = \frac{\max\limits_{i=1}^{m} x_{ij} - x_{ij}}{\max\limits_{i=1}^{m} x_{ij} - \min\limits_{i=1}^{m} x_{ij}} \tag{14b}$$

where $x_{ij}$ and $y_{ij}$ is the original and normalized indicator j in sample i, and m is the total number of samples. Then, the entropy weight method is adopted to calculate SDD, which calculates the information entropy of indicators that reflects their relative change degree on the whole system (Wang et al., 2019). The information entropy of indicator j in sample i is expressed by:

$$E_j = -\frac{1}{\ln m} \sum_{i=1}^{m} d_{ij} \ln d_{ij} \tag{15a}$$

$$d_{ij} = \frac{y_{ij}}{\sum\limits_{i=1}^{m} y_{ij}} \tag{15b}$$

Finally, the entropy weight of each indicator is expressed by:





$$\omega_j = \frac{1-E_j}{\sum_{j=1}^{n}\left(1-E_j\right)} \tag{16}$$

where n is the total number of indicators in a certain module.

The SDD is calculated based on the coupling coordination degree (Sun and Cui, 2018) that reflects the degree of coordination of various factors or subsystems. In this study, SDD is calculated based on the coordination of three modules (EEF) and expressed by:

$$SDD=\sqrt{C_1 C_2} \tag{17a}$$

$$C_1=\left[\frac{ECNMY\left(t\right)\cdot ECLGY\left(t\right)\cdot FOOD\left(t\right)}{\left(ECNMY\left(t\right)+ECLGY\left(t\right)+FOOD\left(t\right)\right)^3}\right]^{\frac{1}{3}}$$

$$=\left[\frac{\sum_{p=1}^{P}\omega_{pj}y_{pj}\cdot\sum_{q=1}^{Q}\omega_{qj}y_{qj}\cdot\sum_{r=1}^{R}\omega_{rj}y_{rj}}{\left(\sum_{p=1}^{P}\omega_{pj}y_{pj}+\sum_{q=1}^{Q}\omega_{qj}y_{qj}+\sum_{r=1}^{R}\omega_{rj}y_{rj}\right)^3}\right]^{\frac{1}{3}} \tag{17b}$$

$$C_2=\frac{1}{3}\left(ECNMY\left(t\right)+ECLGY\left(t\right)+FOOD\left(t\right)\right)$$

$$=\frac{1}{3}\left(\sum_{p=1}^{P}\omega_{pj}y_{pj}+\sum_{q=1}^{Q}\omega_{qj}y_{qj}+\sum_{r=1}^{R}\omega_{rj}y_{rj}\right) \tag{17c}$$

where ECNMY(t), ECLGY(t) and FOOD(t) are the coordination degree of economy, ecology and food module, respectively. P, Q, R is the total indicator number in economy, ecology and food module.

*2.4 Research framework*

      Based on the framework of sustainable development of water resources and the above methodology, the
achievement of sustainability of water resources is solved by coupling multiple objective optimal model and SD model of EEF nexus in the whole process in this research. According to the outline of the EEF nexus, reservoirs are of relatively high robustness for ecological module, while rainfall and evapotranspiration are also indispensable for crop yield equation of food module and development level of socio-economy is also of great significance in economic module. Hence, they should be clarified before implementing the EEF nexus model. The development level of socio-economy is
reflected by the population size and GDP. The observed monthly historical streamflow data, precipitation and temperature can be regarded as the ensemble of reservoir inflow predictions, areal precipitation and potential evapotranspiration in the near future, respectively (Feng et al., 2019). That is, for a certain year in the future, streamflow data of historical decades can be treated as all the possible reservoir inflows, while precipitation and temperature data of the same time scale can be utilized as the input of crop yield equation for all possible hydrological frequencies. It should
be noted that reservoir construction would change the natural streamflow regime and the statistics after construction would be inconsistent with those before construction. Therefore, the streamflow data period after construction should be "restored" to keep consistent with those before construction. The details of restoring methods can be found in Deng et al., (2015). All the optimization results, including water supply and demand, food production, objective functions, can





be obtained to reflect the operational decisions for average level within a particular year. The whole research procedure
is shown as follows:

First, initialize the EEF nexus model. The parameter of the model includes reservoir storage, the water recession rate from previous sub-area, initial streamflow release from reservoir(s), hydrological data (rainfall and evapotranspiration), recycled water usage rate and predicted socio-economic index for each horizon year (initial parameter settings are shown in **Table 3**).

Second, start the optimal model within each horizon year by using the decomposition-coordination principle and dynamic programming. If the optimal water allocation scheme for each year is generated, go to the next step. Otherwise, repeat this step (Section 2.2).

Finally, the optimal solution is used to drive the system dynamic model and simulate the trajectories of corresponding variables of EEF nexus (including objective functions in the optimal model, streamflow water, carrying capacity, food production) and evaluate the coordinative degree of each module and SDD (Section 2.3).

## 3. Case Study

### 3.1 A brief description of study area

Guijiang River Basin (GRB) is one of the most imperative branch basins of the Xijiang River Basin (XRB) in South China. XRB belongs to the typical karst area and is the second-largest river basin in China in terms of total runoff and also the third largest river basin in terms of total area. The names of the mainstream of XRB are Nanpan River, Hongshui River, and Xijiang River in the upper, middle and lower stream, respectively. Yujiang, Liujiang and Guijiang are the main branch river of XRB (see **Fig.5**). The upper reach of Guijiang River Basin (UGRB) (24°6'~25°55'N, 110°~111°20'E) is selected as a case study as it represents the highly conflicts between socioeconomic growth and ecological protection in karst areas. Furthermore, reservoirs are widely constructed in UGRB to supply water for socio-economy but are likely to deteriorate the river ecological health by alternating natural flow (Yin et al., 2010; 2011). UGRB is also a karst area with a total area of 13,131 km$^2$, with a total population of about three million people. Also, UGRB has a total crop planting area of about 2,400 km$^2$, a total vegetation area of about 3,700 km$^2$, and yearly average precipitation of about 1600mm. UGRB is located in Guilin City and refers to eight administrative regions (or counties). Seven reservoirs are constructed in UGRB to provide water resources support for maintaining the development of socio-economy. The detailed parameters of seven reservoirs and their three-level hieratical structure including subareas are found in Supplementary material S4 in Appendix B. Guilin city is both heavy industrial city and national major tourist city, and the population and economic development will keep rapidly increasing in the near future. It will exacerbate the conflicts between social development, food safety and environmental protection especially for that of river ecological environment, resulting in severe ecological deterioration of the lower Guijiang River basin and even lower XRB. Therefore, how to achieve coordination and sustainable development in UGRB between these aspects is becoming a challenging problem in upcoming years and is necessary to be solved.




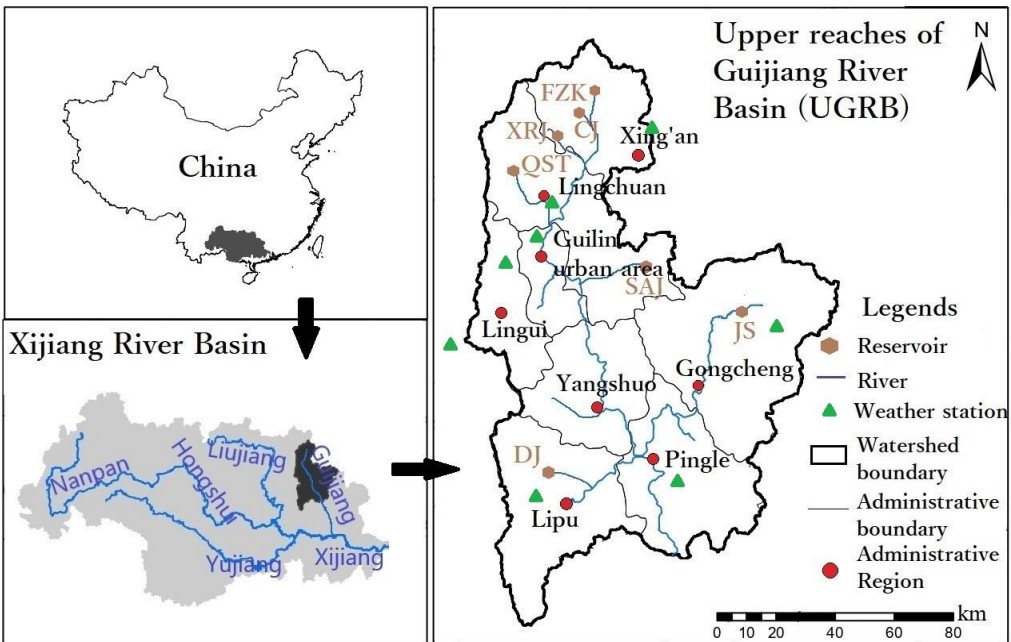

**Fig.5** A brief location of UGRB

*3.2 Datasets and parameter initialization*

Datasets of the case study include socio-economic, water use, land use, meteorological and hydrological data. The major source of socio-economic data, including population and GDP, are the statistical yearbooks of both Guilin City and Guangxi autonomous region from 2005-2014 (http://data.cnki.net). The Municipal Government of Guilin City (MGGC) predicted population and GDP till 2040, along with per capita water use that is from the water industry standard of the People's Republic of China, to predict the water demand of economic module (Venkatesan et al., 2011). The growth
rate is based on these predictions and shown in **Table 3**. The sharply increased rate occurred in the second stage, which corresponds to the era that "heavy government policy support and investment" and "population grow rapidly" as what Kandasamy et al. (2014) stressed in "pendulum model" (see Section 2.1.1). The growth rate from 2031 to 2040 is lower compared with that from 2021 to 2030, which corresponding to the era of "remediation and emergence of the environmental customer" as stated in Kandasamy et al., (2014). Water use data include historical water usage and total
water amount found in Guilin water resources bulletin (2005~2014). Land use data contain the spatial distribution of crops and vegetations with a resolution of 1km×1km that can be found in the Resource and Environment Data Cloud Platform, China Academy of Sciences (REDCP-CAS) (http://www.resdc.cn). Meteorological data from 1956 to 2013, including daily average wind speed, sunshine duration, maximum and minimum temperature, relative humidity and precipitation, are found in meteorological stations (http://data.cma.cn) and are used to calculate $ET_0$ and effective
precipitation that is the main input of crop production equation. The hydrological data from 1958 to 2013 include the monthly inflow of each reservoir and can be found in hydrological stations. All the initialized parameters are list in **Table 3**, and the total index of the data sources can be found in Supplementary materials S5 in Appendix B.

*3.3 Results and discussion*

     The proposed theoretical model is implemented in this case study to acquire the coevolution mechanism and



response link of the EEF nexus. Time scale is divided into three stages: 2016~2020, 2021~2030 and 2031~2040, which correspond to different states of the "pendulum model" addressed by Kandasamy et al. (2014). The proposed study area is also an integrated water resources system with seven reservoirs and eight subareas and can be separated into several subsystems based on section 2.2.1. The detailed conceptualization model of UGRB is presented in Supplementary material S4.


**Table 3** Initial parameter setting of EEF nexus model

| Parameter | Notation | Unit | Eq. | Value | Data sources |
|---|---|---|---|---|---|
| Population growth rate | - | % | (1c) | Stage1: 1.23 | http://data.cnki.net; |
| | | | | Stage2: 3.41 | MGGC; |
| | | | | Stage3: 1.24 | Kandasamy et al.; (2014) |
| Tertiary industrial product growth rate | - | % | (1c) | Stage1: 1.99 | |
| | | | | Stage2: 4.11 | |
| | | | | Stage3: 2.36 | |
| Industrial product growth rate | - | % | (1c) | Stage1: 3.04 | |
| | | | | Stage2: 5.33 | |
| | | | | Stage3: 1.24 | |
| Household water drainage coefficient | $\alpha_1$ | - | (4) | 0.75 | Water resources Bulletin of Guilin City; |
| Industrial water drainage coefficient | $\alpha_2$ | - | (4) | 0.75 | Water resources Bulletin of Guangxi autonomous region |
| Household sewage water treatment rate | $\beta_1$ | % | (4) | 70 | |
| Industrial sewage water treatment rate | $\beta_2$ | % | (4) | 70 | |
| Utilization rate of recycled water | $\zeta$ | % | (4) | 20 | |
| Correction coefficient of soil moisture | $K_s$ | - | (2a)(2b) | 0.9 | Shi et al., (2016); Saxton et al., (1986) |
| Correction coefficient of canopy | $K_c$ | - | (2a)(2b) | Forest: 1.00 | |
| | | | | Open forest: 0.73 | |
| | | | | Shrubbery: 0.65 | |
| Vegetation area | - | km$^2$ | - | Forest: 2373 | http://www.resdc.cn |
| | | | | Open forest: 356 | |
| | | | | Shrubbery: 764.2 | |
| Crop coefficient in different stages | $K_{c,ini}$, $K_{c,mid}$, $K_{c,end}$, | - | (5a) | Rice: 1.05, 1.2, 0.75 | Allen et al., (1998) |
| | | | | Corn: 0.3, 1.2, 0.6 | FAO, 2012 |
| | | | | Vegetables: 0.65, 1.1, 0.95 | |
| Crop area | - | km$^2$ | - | Rice: 1239 | http://www.resdc.cn |
| | | | | Corn: 208.83 | |
| | | | | Vegetables: 670.43 | |
| Initial streamflow of reservoir(s) for monthly average | $Q_{mj}$ | m$^3$/s | (3) | Ecological basic flow, i.e., 30% of average annual flow from April to September, 10% from October to | Hong et al., 2016; Tennant et al., 1976; Hydrological yearbook of |





| | | | | March, based on Tennant method. | Xijiang River Basin (1956~2013) |
|---|---|---|---|---|---|
| Coefficient of big livestock production equation | $a_L$ | - | (7) | 0.002 | Regressive calculation based on Water resources bulletin of Guilin (2005~2014) and Socioeconomic Bureau of Statistics of Guilin City (2005~2014). |
| Coefficient of big livestock production equation | $b_L$ | - | (7) | 0.0405 | |
| Coefficient of poultry production equation | $a_L$ | - | (7) | 0.0028 | |
| Coefficient of poultry production equation | $b_L$ | - | (7) | 0.00002 | |

*3.3.1 SD model calibration and validation*

The historical data was used in the model for calibration and validation by comparing simulated and historical results. Some parameters, including sewage water treatment rate, water drainage coefficient, the utilization rate of recycled water, should be calibrated before validation and is also shown in **Table 3**. The yearly comparison of simulated
and historical household & industrial, agricultural and off-stream vegetation water use is shown in **Table 4**. They are corresponding to the economic, food and ecology module of EEF nexus model. We can see that simulation results of the SD model are well matched with the actual values, with the relative error of less than ±5%, indicating that the proposed model is reliable and can simulate the future coevolution process.

**Table 4** Comparison of water usages over the years

| Year | Household & industrial water usage ($10^8 m^3$) | | | Agricultural water usage ($10^8 m^3$) | | | Vegetation water usage ($10^8 m^3$) | | |
|---|---|---|---|---|---|---|---|---|---|
| | Simulated | Actual | Relative error | Simulated | Actual | Relative error | Simulated | Actual | Relative error |
| 2005 | 4.07 | 4.23 | 3.93% | 12.56 | 12.23 | -2.63% | 21.08 | 21.55 | 2.23% |
| 2006 | 4.18 | 4.08 | -2.39% | 12.45 | 12.13 | -2.57% | 20.67 | 20.98 | 1.50% |
| 2007 | 4.34 | 4.46 | 2.76% | 12.26 | 12.54 | 2.28% | 20.89 | 21.44 | 2.63% |
| 2008 | 4.13 | 4.22 | 2.18% | 12.65 | 12.35 | 2.37% | 21.23 | 21.77 | 2.54% |
| 2009 | 4.02 | 4.10 | 1.99% | 12.56 | 12.76 | 1.59% | 20.45 | 21.23 | 3.81% |
| 2010 | 4.15 | 4.09 | -1.45% | 12.43 | 12.77 | 2.74% | 21.45 | 20.89 | -2.61% |
| 2011 | 4.11 | 4.13 | 0.49% | 12.44 | 12.67 | 1.85% | 21.23 | 22.24 | 4.76% |
| 2012 | 4.23 | 4.16 | -1.65% | 12.36 | 12.78 | 3.40% | 21.11 | 21.89 | 3.69% |
| 2013 | 4.18 | 4.21 | 0.72% | 12.45 | 12.65 | 1.61% | 20.68 | 21.12 | 2.13% |
| 2014 | 4.28 | 4.19 | -2.10% | 12.37 | 12.54 | 1.37% | 21.56 | 20.98 | -2.69% |

*3.3.2 Coevolution process of EEF nexus*

The coevolution trajectories of population, GDP, water supply & demand, streamflow and objective function ($F_{ecnmy}$, $F_{eclgy}$, $F_{food}$) (based on Eq.(8),(9),(10)) referring to each component of the EEF nexus is shown in **Fig.6**. As can be seen in **Fig.6**, the coevolution process of all the items shows the characteristics of different stages. Finally, the (quasi-)stable state is converged, i.e., the variations of each variable are small or close to zero. It happens because the rate of external
changes in the last stage (i.e., economic indexes) is much lower than in the previous stage, which decreases the internal changes (i.e., Streamflow water and three objective functions) and finally the stable status of the whole system is achieved. In the first stage, the growth rate is relatively low and is based on the historical data, and the growth rate of $F_{ecnmy}$, $F_{eclgy}$ and $F_{food}$ is also slow. When entering the second stage, the economic growth will increase sharply to ensure the local economic development, and water demand is also increasing. However, according to the achievement of
sustainable development based on the optimal model, ecological concerns should not be neglected. Therefore, the increase of river streamflow will also happen driven by the optimal model to maintain the river ecological health,





consequently reducing the total water supply and increasing the water shortage of water users (**Fig.6**c). As $F_{food}$ and $F_{ecnmy}$ can reflect the water shortage of the corresponding water users, their value will also increase sharply (**Fig.6e** and **6g**) due to the rapid increase of socio-economic indexes. When entering the last stage, the development of socio-economy

will tend to stable, and the increasing speed of $F_{food}$ and $F_{ecnmy}$ will decrease compared with that in the second stage. It is easy to understand because the relatively stable development of socio-economy does not need too much increased streamflow water (the increase rate of streamflow water is also closed to a relatively stable state), and both changing rates of water supply and demand tend to be stable consequently (**Fig.6**c).

We can also see that the water supply system competes for the instream ecological system. As shown in **Fig.6**,

especially in stage 2, increased streamflow is accompanied by increased $F_{ecnmy}$ and $F_{food}$ (**Fig.6e** and **6g**), reflecting the decreased satisfaction degree of the water supply of socio-economy and agriculture, thereby revealing the competition use of instream and off-stream water uses. The trade-off between instream and off-stream water users can be obtained by the optimal model to solve for the best coordination status between them by adjusting economic development modes and balance the priority of each water users. It should be noted that the ecological objective ($F_{eclgy}$) is in a relatively

stable status in all stages compared with other objectives (**Fig.6**f). This is because the ecological module contains not only river streamflow but also vegetation. The booming economy drives the optimal model to focus more on river ecological health ($F_{riv}$) and there are limited water resources for off-stream water users including vegetation. The dual effect of increasing streamflow water and decreasing water for vegetation makes the $F_{eclgy}$ relatively stable. However, the optimal model takes the effect that the optimal allocation scheme is obtained by shifting streamflow water because

instream and off-stream water use is intrinsically conflicted with each other, and should be coordinated by adjusting different weights of each component (see section 3.3.5).



**Fig.6** Coevolution process of EEF nexus model

*3.3.3 Performances of feedback loops and response linkages*

**Fig.7** illustrated the loop of economy-ecology feedback. As demonstrated in **Fig.7**, the response linkage of carrying capacity and overload index involves the changes of economic indexes, water supply & demand and streamflow water (Feng et al., 2019). In the beginning, the economy is still increasing slowly, and the increasing rate of water demand is also slow. The population and GDP are near the carrying capacity in this stage (i.e., the value of OI is near 1). In the following stage, both increasing population and GDP intensify the water demand (**Fig.7a** and **b**). To satisfy

socioeconomic development demands, water supply of economic module has also increased. However, according to the coevolution of the whole system obtained by optimal model, there will be a higher concern of the river ecological system



(**Fig.6**c, **Fig.7**c). In this view, the feedback linkage will take effect as that the growing rate of water supply of household and industry (**Fig.7**d) will fail to catch the rate of water demand (**Fig.7**b) and therefore contributes to the increase of water shortage, which is in accordance with the performance shown in **Fig.6**e. The increasing water shortage will generate the gap between carrying capacity (**Fig.7**e) and predicted economic indexes (**Fig.7**a). Then, the overload index will further increase, consequently affecting the socio-economic development. It will force the local policymakers to readjust the regional development level and influence the population and GDP, indicating a new round of feedback. In this view, we can see that the rapid growth of economy in the second stage will activate the protection mechanism of river ecology by increasing the streamflow, and the rest water is unable to support the increasing economic development. It further contributes to the overload of the water resources system, which even restricts the socio-economy instead. In the last stage, the continuous increase of the overload index stimulates the policymakers to alleviate the increase rate of population and GDP (**Fig.7a** and **f**). It forces the relatively slower increase rate of streamflow water and there will be more water space for socio-economic development. Although the water shortage is increasing, its rate is lower than that in the second stage. The carrying capacity will be able to catch the predicted economic index if the stable or slower growth rate continues. The overload index is also decreased and the whole system tends to be stable.

**Fig.7** Response linkage of economy-ecology feedback loop

Another performance is the ecology-food response linkage and is shown in **Fig.8**. It not only illustrated the linkage between food and ecological water usage but also demonstrated the coevolution of ecology components of both instream (river ecology) and off-stream (vegetation) aspects. We can see from **Fig.8** that the increased streamflow water is the driving force of ecology-food response. However, the increasing streamflow water was driven by the rapidly increasing socio-economic scale. The optimal model is used to achieve the goal of sustainable development to balance the need of different users, especially that of instream and off-stream. The increased streamflow has two effects in ecology-food response linkage. First, the variable $F_{riv}$ describes the ecological health of a certain river. According to Eq.(3) and Eq.(9c), the higher value of streamflow water indicates the lower value of $F_{riv}$, which indicates that the river ecology is getting better. Second, the increasing streamflow water restricts the water supply of all off-stream water users, including agricultural and vegetation water (**Fig.8**b). Irrigation and vegetation water use is the largest off-stream water consumer, and their increased water shortage was also driven by increased streamflow water (**Fig.8**d). It should be noted that the food module includes not only crops but also livestock. Livestock breeding will inevitably increase to



make more production value of primary industry, and there will consequently be more water demand for livestock.



**Fig.8** Ecology-food response linkage

The dual effect of increased streamflow water and decreased vegetation water makes the stable change of $F_{eclgy}$ (**Fig.8**e), indicating that the ecological aspect of UGRB is maintaining a good status. According to Eq.(6), crop yield is

strongly affected by the satisfaction degree of irrigation water, and the increased water shortage of crop water will, therefore, indicate the decrease of crop yields (**Fig.8**f). In contrast, the decreased water shortage of livestock could induce an increase in meat production. The detailed changes of crop yield and meat production are presented in **Fig.9**. We can see from **Fig.9** that a relatively large proportion of food production is from crop yield. Although meat production is increasing, it accounts for relatively less proportion, and thereby the total food production will first decrease and then

tend to be stable in the last stage (**Fig.9**c). Besides, the decreased food production is driven by the increased streamflow water that also caused an increasing overload index (**Fig.7**f) especially in the second stage. Thus, we can infer that the decreased food production may also indirectly increase overload index, and is verified by the demonstration of




comparison between **Fig.9**c and **Fig.7**f in the second stage. Simultaneously, it is clear that the relatively stable changing rate of food production occurred in the last stage (**Fig.9**c), accompanied by the decreased overload index in the same

period (**Fig.7**f). This is the economy-food response linkage that takes effect as the higher socioeconomic growing rate will have an adverse effect on food safety, further affecting the carrying capacity. Therefore, the linkage of economy-food, economy-ecology and ecology-food were all presented, which indicated that the three components interact and respond with each other.

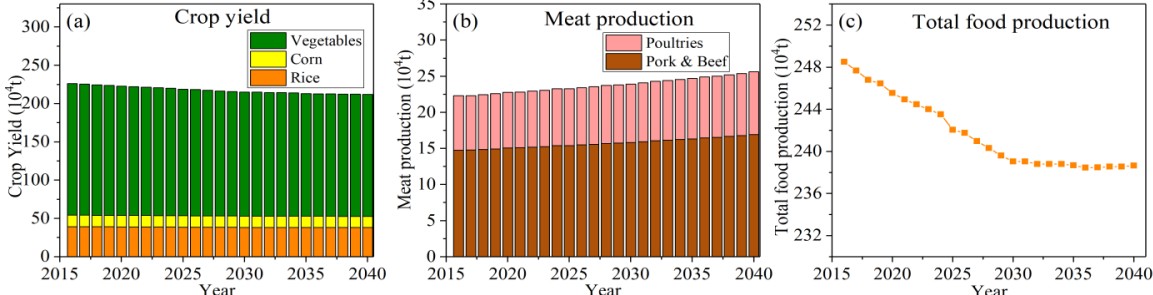

**Fig.9**   Detailed crop yield and meat production in the next 25 years in UGRB

*3.3.4 Assessment of coordinative degree of each subsystem and SDD*

    The calculation result of SDD of EEF nexus and coordination degree of economy (ECNMY), ecology (ECLGY) and food is demonstrated in **Fig.10**. We can see that the variation of the four variables is also showing the state characteristics. The ECNMY in the first stage is increasing, but it had an either decreasing (UGRB, Guilin urban area,

Lingui, etc.) or stable (Xing'an, Yangshuo) trend in the second stage, indicating the coordinative status of socio-economy is not good caused by the excessive growth rate of economy. The decreased coordinative status of economy subsystem also influences other subsystems and the SDD of total EEF nexus, reflected by the decrease of ECLGY, FOOD, and further SDD. Fortunately, the decreasing rate of ECLGY is smoother compared with that of FOOD, indicating the performance of ecology of UGRB is relative well compared with socioeconomics and agriculture. This performance

could be due to the dual effect of increasing streamflow water, sewage and recycled water treatment, and decreasing vegetation irrigation. The same was true for other administrative regions of UGRB. Moreover, for the whole basin, the value of ECNMY in the later period of the second stage (about 2028~2030) is even lower than FOOD and ECLGY, From the perspective of administrative regions, it is more obvious in Guilin urban area, Pingle and Lipu counties. It happens because the economic-stressed stage has been last almost ten years in 2030, which is similar to the "pendulum model"

that takes the effect that the pendulum "swing" towards the economic-stressed system (See 2.1.1). As socio-economic index increases sharply and continuously, the ecological protection mechanism will also be continuously triggered to increase the overload index, resulting in both ECNMY and SDD reached the minimum.

    When it comes to the third stage, the value of ECNMY increases, indicating the coordination of the economic subsystem is improving. It revealed the decreasing of overload index and the increasing carrying capacity, due to the

relatively slower increasing rate of water demand of economic module. The increasing value of ECNMY even promotes the coordinative degree of ecology and food, and the value of SDD is consequently increased, revealing that the stable economic growth will promote the sustainable development of EEF nexus. The good phenomenon of the last stage happens because the relatively slow growth rate of water demand for the economic module will generate more water for food and ecology, and the increasing sewage and recycled water treatment rate will provide relatively more water for

users. The coevolution process is based on the assumption of the "pendulum model" presented by Van et al., (2014) and Kandasamy et al., (2014), where the environmental awareness has raised, and stable population rate occurred in the last era. The result presented in this study is similar to the findings in Van et al., (2014) and Kandasamy et al., (2014). Furthermore, we can speculate that in the 2040s, the pendulum of ULRB will also "swung" back to the stage of protective





resources & environment and stable development of socio-economy, just as stated in Kandasamy et al., (2014).

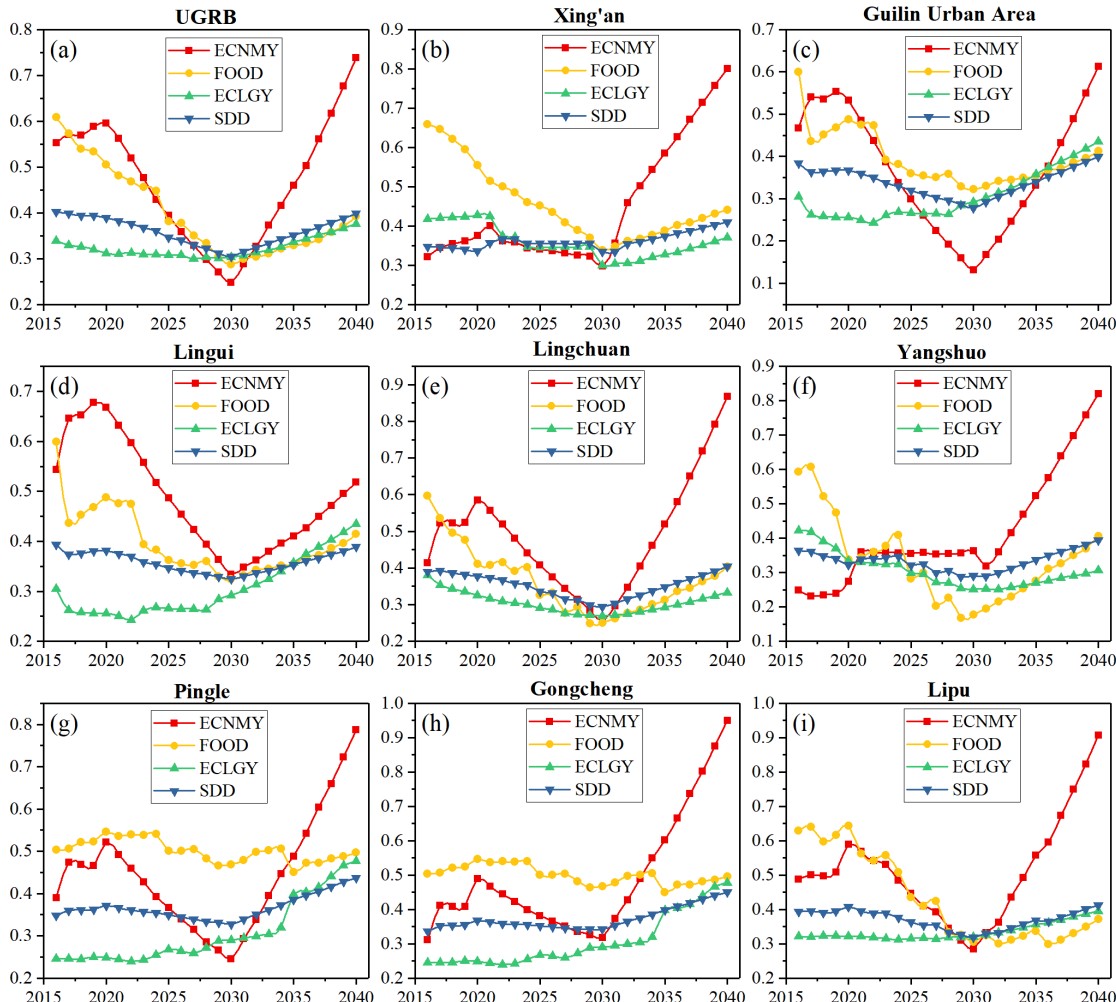

**Fig.10** Time variation of sustainable development degree (SDD) of EEF nexus and coordination degree of each module

*3.3.5 Sensitivity analysis of decision preferences considering weight uncertainty*

In the current study, the optimal decisions over the years were obtained by optimal model. Each water user is set equal to take into account each stakeholder to fully achieve the sustainable development. However, due to the internal conditions of different regions and preferences of policymakers, the weight of each component is usually different and difficult to be determined since the equal weight for each water users for sustainable development may not applicable in every regions or watersheds, which is one of the most important sources of uncertainties. Since the most contradictory among water users is the conflict between instream river ecology and off-stream water users (Homa et al., 2005; Yin et al., 2010; Shiau and Wu, 2013; Rheinheimer et al., 2016), the uncertainty is mainly embedded in different values of θ (See Eq.(12c)) that reflects the priority of streamflow water, which is also the main variables of coevolution process. The results of uncertainty and sensitivity analysis is shown in **Fig.11**.





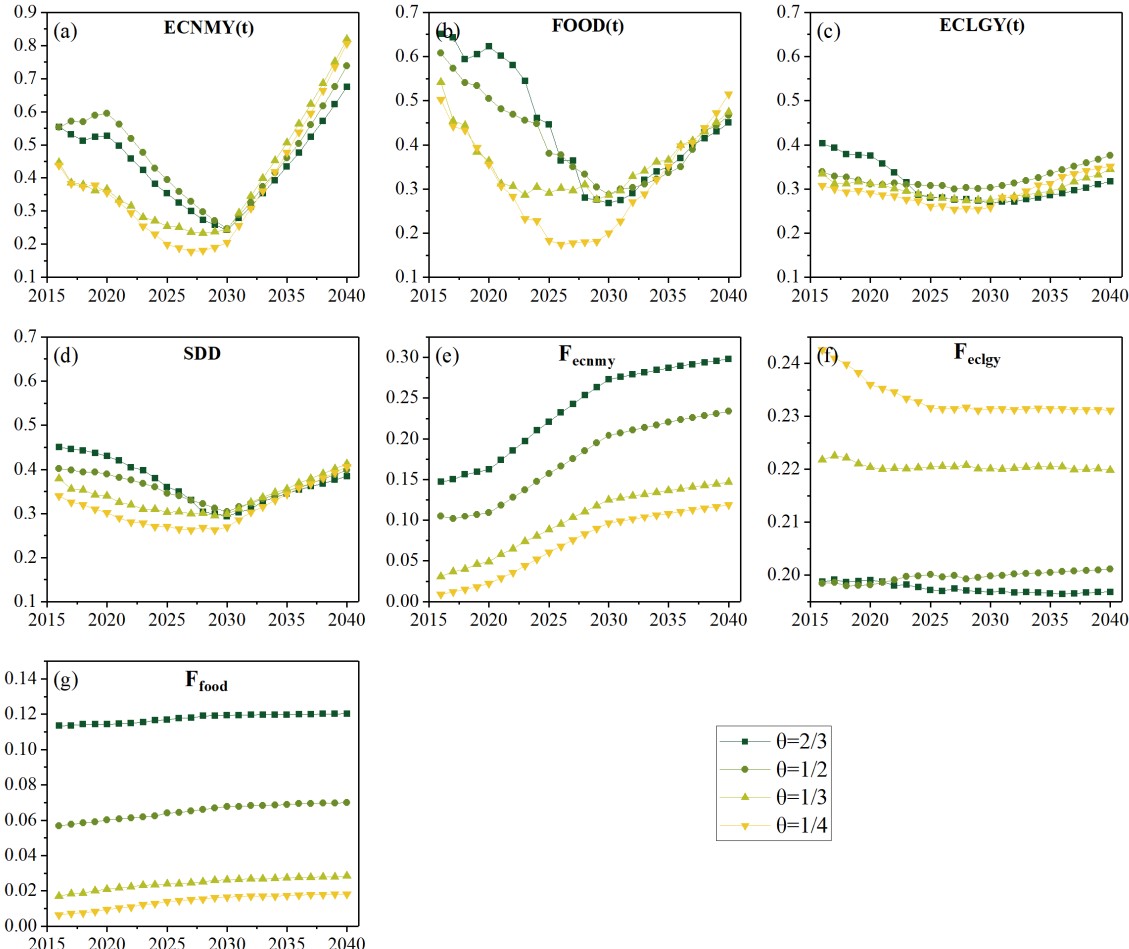

**Fig.11** Sensitive analysis of trajectories of coevolution, coordination and sustainable development degree under uncertain decision preferences

640       According to Eq.(12), the more value of $\theta$ is, the more concern of instream water uses. As demonstrated in **Fig.11**, the coordination degree of each component and their sustainable development (SDD) shows the different sensibilities from stage perspective, especially that of ECNMY(t), FOOD(t) and SDD. We can see that these three variables under the different value of $\theta$ in the first two stages are sensitive, but they converge to a similar trend in the last stage. That is, the difference of these variables under different value of $\theta$ in the last stage is small compared with that in the second

stage, indicating that the decision preference is sensitive to $\theta$ when the growth rate of socio-economic index is exceptionally high, while the trajectories of these valuables are insensitive to $\theta$ when the growth rate of population size and GDP are relatively stable.

      When it comes to ECLGY(t), it shows no apparent sensitivities because the ecological module contains river and vegetation ecology that belong to the instream and off-stream water use category. However, the different weight of

streamflow water $\theta$ gives the guideline for ecological stakeholders. The higher value of $\theta$ indicates a lower value of vegetation weight ($\alpha_3$) (See Eq.(12)) as they are of different categories (instream and off-stream) and conflict with each other. Although they both belong to ecological water usage, they also should be coordinated through different weighting factors ($\theta$ and $\alpha_3$). The deeper concern of streamflow water should be attracted (with a higher value of $\theta$ of 2/3) in 2020s





to increase ECLGY as off-stream water users are increasing dramatically. In comparison, off-stream water demands tend
to be stable in 2030s and vegetation concern can be moderately increased (with a lower value of θ of 1/2) to increase the
value of ECLGY. However, the shallow value of θ (1/3 or 1/4) cannot perform the best for ecological module as
streamflow water cannot be allocated to an extremely low weight. The balance of both ecological water usage is still
needed. In addition, of all these four assessment variables, the trend in the second stage is decreasing while it is increasing
in the last stage, showing that the relatively stable growth of socio-economy promotes the coordination of each
component and the sustainable development among EEF nexus, instead of dramatical growth.

As different values of θ can influence the performance of EEF nexus especially in the second stage in which higher
sensitivity shows, the result can also give a reference to policymakers for tradeoffs of more than just ecological
stakeholders. We can see that in the stage on which the higher emphasis of the economy was put (2020s), if the ecological
awareness was still neglected (θ=1/3 or 1/4), there will be less coordination degree of economy (ECNMY) and food
(FOOD) as well as SDD (**Fig.11**a,b and d). Therefore, the tradeoffs between instream and off-stream water usage should
be fulfilled to achieve the coordination and sustainable development, i.e., in the stage to which the economic aspect is
paid high attention, the weight of river ecology should be inversely set higher in the objective function of the optimal
model to achieve the relative equilibrium between instream and off-stream water uses. To prove, we can see that the
performance of ECNMY(t), FOOD(t) and SDD is better under θ=2/3 or 1/2 compared with the condition when θ=1/3 or
1/4. But the performance of coordination and sustainable development would no longer be that sensitive when the
economic growth rate is stable.

The preference of θ also influences the objective functions of the EEF nexus model. As can be seen from
**Fig.11**e~11g, a higher value of θ results in lower $F_{ecnmy}$ and $F_{food}$ and higher $F_{eclgy}$, indicating the smaller water shortage
of economic and food modules and less awareness of ecology, and vice versa. However, only by evaluating the objective
functions is too one-sided to reveal the interaction of EEF nexus and cannot give a comprehensive reference for water
resources management. It should still be couple with the coordination degree and SDD to give the reference for
policymakers on the full scale.

## 4. Conclusion

In this study, we incorporated the coevolution process between EEF nexus systems by coupling a theoretical
framework with a system dynamic model. The coevolution model contributes to explore the dynamic changes of each
system as well as the status of sustainable development of water resources within the EEF nexus framework. The multiple
objective model is used to obtain the optimal water allocation scheme of each system, while system dynamic model is
used to explore the dynamic coevolution process and response linkages, as well as sustainable development degrees
across these components. The coupled model was applied in the upper reaches of Guijiang River Basin, China, and the
sensitivity and uncertainty analysis are also conducted to better understand the model performance. The following
conclusions can be drawn from this study:

(1) The coupling models can efficiently reveal the water allocation scheme, coevolution process, optimal decisions
and tradeoffs under the changing external conditions. The feedback loops and response linkages (including economy-
ecology, economy-food and ecology-food) can be mathematically expressed, providing a powerful tool for better
understanding the constitutive linkages and properties of EEF nexus.

(2) The changes of socio-economic indexes will result in the shifting in behaviors of the optimal water allocation
scheme, i.e., the rapid socioeconomic development will raise the awareness of environmental protection, reflected by
the increasing of reservoir streamflow, and further influences the dynamic performances and coevolution of other
components. The excessive socioeconomical development will trigger the ecological protection mechanism, further
increasing water shortage, and decreasing food production. Furthermore, these changes will increase the population
vulnerability and overload index, which will also vulnerable to sustainable development degree, and ultimately have a





bad impact on the coupled system instead. Once the socioeconomical growth rate is stable, the coordination and sustainable development degree of the whole system will be improved.

(3) Sensitivity and uncertainty analysis revealed that different preferences can influence the coevolution process and the status of coordination and sustainable development. Its performance is sensitive to the awareness of reservoir streamflow when the growth rate of socio-economy is extremely high. In this case, more emphasis of streamflow should be put on to improve the coordination of each component and sustainable development across the subsystems. The coordination and sustainable development degree are insensitive to the reservoir streamflow under the lower growth rate of socio-economy.

The proposed model is adopted in a case study in a typical karst area in South China, and the results present in this study can give powerful references for decision makers to identify the coordinated management and assess comprehensive plans. The theoretical framework and methodology presented in this study are suitable for any other watersheds and regions that contain reservoirs, especially for areas with prominent conflicts between multiple water users. Although this study attempts to present a new framework of economy-ecology-food nexus, there is still room for

improvement. For example, sustainable development also contains energy, land use, climate change and other aspects, which consequently increases the dimension of the model and more complicated approaches might be introduced to obtain the optimal solutions. Moreover, the sources of uncertainties not only contain the conflicts between instream and off-stream water usages but also the usages within off-stream, probably reflected by the different structures of industries, which will also be our further works of the future research.

**Acknowledgements**: The project was financially supported by National Key Research and Development Program of China (No. 2018YFC1508200), National Science Foundation of Jiangsu (No. BK20181059) and China Scholarship Council. The authors were also grateful to the sources of hydrological and meteorological data from hydrological authority and statistical bureau, and the organizations and comments handled by Dr. Zengchuan Dong and Dr. Sandra M. Guzman. The authors are still grateful to the insights and views of the editors and reviewers.

**Appendix A: Main variable index of EEF nexus**

| Variables | Description |
| --- | --- |
| $WD_{hou}$ | Household water demand |
| $WD_{indus}$ | Industrial water demand |
| $q_{hou}$ | Household water use quota |
| $q_{indus}$ | Industrial water use quota |
| $WD_{veg}$ | Vegetation water demand |
| AAPFD | Amended annual proportional flow deviation, which is used to assess river ecological health in ecological module |
| $F_{eclgy}$ | Objective function of ecological module |
| $F_{ecmny}$ | Objective function of economic module |
| $F_{food}$ | Objective function of food module |
| $F_{riv}$ | Objective function of river ecological health |
| $F_{veg}$ | Objective function of vegetation water use |
| $WD_{jkt}$ | Water demand of jth water use sector of kth subarea at tth time period |
| $WS_{jkt}$ | Water supply of jth water use sector of kth subarea at tth time period. |
| $WD_{ecmny,kt}$ | Water demand of economic module of kth subarea at tth time period |
| $WS_{ecnmy,kt}$ | Water supply of economic module of kth subarea at tth time period |
| $WD_{food,kt}$ | Water demand of food module of kth subarea at tth time period |
| $WS_{food,kt}$ | Water supply of food module of kth subarea at tth time period |





| WD$_{veg,kt}$ | Water demand of vegetation of kth subarea at tth time period |
| WS$_{veg,kt}$ | Water supply of vegetation module of kth subarea at tth time period |
| Y$_a$ | Actual crop yield |
| Y$_p$ | Ideal crop yield |
| Y$_L$ | Livestock (meat) production |

## Appendix B: Supplementary materials (Data availability)

### Author contribution

Yaogeng Tan prepared the manuscript and developed the model. Zengchuan Dong and Sandra M Guzman revised the manuscript. Sandra M Guzman also helped developing the model. Xinkui Wang and Wei Yan helped collected the data.

### Competing interests

The authors declare that they have no conflict of interest.

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
