# Peer review of "An integrated modeling framework for coevolution and feedback loops of nexus"

_Hydrology and Earth System Sciences, 2020_

## Referee Comment (RC1) · Anonymous Referee #1 · 10 Jul 2020

The first problem of developing integrated models is that you need to sacrifice detail somewhere. And in this model, the economic module is clearly not adequate. For example, farmers maximize yield instead of profit/utility, and moreover appear in the food and not the economic module. This means that farmer's response has no economic basis/rationale. As noted in the paper, the goal is simply to maximize Crop yield and meat production. This is by no means the objective of farmers in real life. It is admissible to assume farmers maximize profit (not really, but let it pass). But assuming farmers maximize crop yield will lead to misleading simulation results and policy

recommendations, because they simply don't do that. In my opinion this assumption already makes the model deeply flawed. Agriculture is the largest water user worldwide and you are assuming an objective function that is unrealistic. For example if you have water scarcity in the future you may expect farmers to shift to less water intensive crops / deficit irrigation / rainfed while maximizing profit; which may conflict with the way crop maximizing farmers respond.

The second problem is that in order to combine the models into a single optimization problem / target / goal, the conflicting objectives of each module need to be simplified and this requires some additional assumptions that may not work well; in this case for example, the overall objective of sustainable development looks to me rather normative, which does not really fit into the positive nature of some of the disciplines considered. As an example, authors note that non-linearities are important to accurately represent the EEF nexus but then, within each one of the modules, the objective function is linear, contradicting elementary theory. Assuming economic agents will increase their demand along with population and quotas independently of quality of life, for example, is a simplistic assumption. This may be acceptable as long as household demand is not very much significant in terms of overall demand. But in the agricultural sector, a linear demand that disregards economic incentives and simply looks at the surface of crops and measures ET to assess demand is likely to be inaccurate. What is the rationale for having a mixed portfolio of crops if this is the objective? Or for prioritizing one crop over another one? This is not done in a kg/ha basis, but on a profit basis. The upshot is that while the model may represent well the observed behavior in the year 0, projections are difficult to believe, simply because the key behavioral drivers identified by theory (profit, risk aversion) are missing in the model.

The dynamics between systems have some merit, but they build on a conceptual approach that is inherently wrong. I commend authors to read the paper by (Pindyck, 2015), who warns against over-ambitious and overcomplicated integrated models that for the sake of reproducing complex interactions need to simplify or straightaway ignore

some basic principles and therefore may become "useless as a policy tool".

I am well aware this sort of simplifications and objective functions are often assumed in several models, even if they disregard basic theory in several disciplines. But if the authors want to use this approach it has to be much better justified, and the caveats above need to be highlighted and acknowledged as a critical limitation of the model. To me the useful contribution of the model is the way feedbacks are approached and represented, but the modules are flawed. Therefore any forecast should be done with this important limitation in mind. Right now authors do not warn about this, which may be misleading. If the paper is accepted, such limitations need to be acknowledged.

Connected to the conceptual framework adopted, authors do not deal with the issue of modeling uncertainty. As noted above, authors had to do many several assumptions on model design (and therefore on real life interactions within and between systems) that may be (and in some cases are) wrong. This means the model, and not only scenarios, may be responsible of sim and prediction errors. I'm not talking about calibration residuals, which in the normative model adopted are missing; I'm referring to the error inherent to the model choice. See e.g. (Herman et al., 2015; Marchau et al., 2019)

Minor comments

Please check the English grammar, there are some problems throughout the text. For example, the first statement ("As global warming causedby climate change and growing population, the world isfacing the disequilibrium between natural resources sustainabilityand human wellbeing") appears incomplete.

Herman, J.D., Reed, Zeff, H.B., Characklis, G.W., 2015. How Should Robustness Be Defined for Water Systems Planning under Change? J. Water Resour. Plan. Manag. 141, 04015012. https://doi.org/10.1061/(ASCE)WR.1943-5452.0000509 Marchau, V.A.W.J., Walker, W.E., Bloemen, P., Popper, S.W., 2019. Decision Making under Deep Uncertainty: From Theory to Practice, 2019th ed. Springer, Cham, Switzerland. Pindyck, R.S., 2015. The Use and Misuse of Models for Climate Policy (Working Paper

No. 21097). National Bureau of Economic Research.

---

## Author Comment (AC1) · 12 Jul 2020

Dear referee,

Thank you for supporting the decent contributions of this paper. Also, I'm greatly appreciated for your comments on our manuscript because it can substantially help improve the quality of the paper. I carefully read your comments about this paper and they can, in my opinion, categorized into five concerns. In short, it refers to (i) the economic module of the nexus should contain the farmer's profits; (ii) optimal model development; (iii)

the limitation of the development model that should be acknowledged; (iv) uncertainty of the conceptual framework modeling; (v) English grammar. These main concerns you mentioned are very useful for the paper as well as my future research careers, and I will carefully revise the paper to improve its quality.

Below please find my responses to your comments and the revised directions of the manuscript that we are going to resubmit.

(i) the economic module of the nexus should contain the farmer's profits (corresponding to the first paragraph of your comment)

I completely agree with your opinion because the agricultural economy is also of great significance to socio development and it is reflected by the profit of farmers and primary industry products. This is what the paper lacks and we will add it in the revised paper. However, the source of agricultural profit is still crops and meat production, which is an important part of the agricultural module. It is undoubtedly that humans cannot survive without crops. Generally speaking, crops and meat are usually sold in the farmer's markets and they get profit by customers buying those crops and meat to simulating consumption (just take an example). Therefore, the maximizing profit of farmers, as well as primary industry products are still based on crops.

Of course, if the model only considers the crop yield in the agricultural sector, it is definitely unrealistic as you mentioned in your comment because the farmer's response has no economic basis. Crop is important, but it is not the only. The farmer's profit, as well as primary industry product, should be fully considered as it is an important part of the economic module that the current paper lacks. We will consider this part of analysis into both the model setup and SDD assessment and add it to the model when revising the paper, and better demonstrate the economy behaviors by revealing its mutual interaction between other modules/components to make the model more integrated and realistic, and further adjust the demonstration of feedback loops and results.

(ii) Optimal model development (corresponding to the second paragraph of your comment)

The paper developed the optimal model to achieve the goal of sustainable development, considering three aspects: ecology, economy, and food. The main part of this research is water resources, and an adequate supply of water resources is the key insurance of social activities. Although each objective function seems to be linear, when combining to an integrated model with the optimal programming solution, as well as the system dynamic model that is also complex and nonlinear, in my opinion, the total method may appear to be nonlinear. Moreover, the uncertainty analysis (see (iv)) will also be considered when revising the paper, which will strengthen the model reliability. Agricultural (crop) demand mainly contains irrigation that is mainly determined by precipitation that is random and cannot be determined by human being, which is different from household and industrial water demand. Also, economic incentives are also key sources of agricultural water demand. I agree with your opinion about the nonlinear assumption. For the concern of your agricultural sector, the farmer's profit will be considered in the model to address the profit basis (as stated in (i)).

(iii) The limitation of the development model that should be acknowledged (corresponding to the 3rd and 4th paragraph of your comment)

I have read thoroughly about the reference you offered: Pindyck 2015, and indeed the statement he/she argued is reasonable. Pindyck stated the IAM modeling limitations and took the relationship between climate change and GDP as an example. The main limitation is the flaws of the models that sometimes must be developed to overcome a certain problem. The main source of the flaws is usually the uncertainties of a certain modeling approach that stems from the definitions and assumptions of certain equations, concepts, or parameters that are usually diverse from different researchers. The author took the examples of (i) damage function of DICE (Dynamic Integrated Climate and Economy) model that has both quadratic and cubic assumptions; (ii) "Limits to Growth" debate that slow population growth to prevent the natural resources from running out, which is likely to ignore the basic economics that the price of natural resources will rise to decline their uses. Therefore, the author stated that the limitations of a certain modeling approach should be acknowledged before adopting the model to prevent it from abusing, but he/she did not completely deny the application of the model. For example. IAM model can be valuable as analytical and pedagogical devices to help us better understand climate dynamics and climate–economy interactions, as well as some of the uncertainties involved.

Similarly, the above statement can be also adopted in my study. It is rather necessary to assess the feedback loop and future changes to better understand the achievements of sustainable development but, we are unable to know what will happen in the next one, two or more decades unless we are prophets. Therefore, the "modeling" approach is emerged to "simulate" the possible changes in the future and many researches have also developed such models to simulate future trends (Feng et al., 2016; 2019). However, it certainly refers to somewhat unreliable and uncertainties that stem from the difficulty of being a common sense to a certain issue or concept (as stated by Pindyck that the assessment of social cost of carbon (SSC) is quite different ($39, $11 even $200), which increases the uncertainties of the model, and the different values of SCC are from different scholars). Although it cannot give a policy direction as is usually need to simplify the basic principles, it does not mean that it is of no use, at least it can help us better understand how to achieve the goal of sustainable development. It doesn't mean to give up entirely on estimating the sustainable development status more generally (Pindyck, 2015). Therefore, we need to take advantage of the positive effect as much as possible of a certain model that is, although, usually double-sided. To consider the limitation of the modeling approach and revise it in my paper, I will add the above statement at the end of "result and discussion" section to highlight the limitation to discuss in-depth about the paper to better acknowledge the limitations to readers in the revised paper to prevent misleading.

(iv) Uncertainty of the conceptual framework modeling (corresponding to your last para-

graph to your comment)

As stated before, models are usually a flaw because it contains uncertainty (Pindyck, 2015) and may be responsible for simulation errors. Therefore, as you mentioned in the comments, uncertainty analysis is necessary to improve the reliability of the model. I agree with your suggestion and have read the references you provided to me. According to Herman et al., (2015), many methods were tackled to deal with the model uncertainties. For multiobjective models, Herman stated, "the pre-specification of alternatives, preference weighting, the most important (sensitive) factors, or performance thresholds may produce unintended consequences for decision making." In other words, preference weighting is one of the most important sources of uncertainty. Also, Kasprzyk et al., (2013) stated that MORDM approach is also a method to deal with the uncertainties, in which alternatives are first generated by multiobjective search prior to their evaluation in deeply uncertain states of the world. The alternatives are generated by LHS sample and Pareto frontiers for tradeoffs between objectives. Also, Liu et al., (2019) also started the multiobjective model refers to the wind-photovoltaic-hydropower system that corresponds to the three objectives, and uncertainties are also referred by the different weights of objectives that are addressed by Pareto frontiers.

If uncertainties are analyzed on a full scale in this paper, it is clearly beyond the scope of this paper as uncertainty itself can be written by another paper, which is another story. Also, the paper will be too long. However, uncertainty is of great necessity to improve the model's reliability and therefore, must be considered. In this paper, the EEF nexus model includes three objectives: economy, ecology and food, and uncertainty can be also reflected by weighting factors and tradeoffs between objectives (Liu et al., 2019). Actually, in the current paper, I mentioned this issue in Section 2.2.3 but maybe it's not so clear. Therefore, in the revised manuscript, I will strengthen the statement of uncertainty and tradeoff issue, and the mutual relationship & tradeoff between objectives will be further analyzed to improve the model reliability, which gives more references for policymakers.

[Figure]

(v) English Grammar (minor comments) I will read the manuscript thoroughly and check the grammar everywhere in the revised paper as much as possible.

Yours, sincerely,

Yaogeng Tan and co-authors

References

Feng, M., Liu, P., Guo, S., David, J. Y., Cheng, L., Yang, G., & Xie, A.: Adapting reservoir operations to the nexus across water supply, power generation, and environment systems: An explanatory tool for policymakers. J Hydrol, 574, 257-275, 2019.

Feng, M., Liu, P., Li, Z., Zhang, J., Liu, D., & Xiong, L.: Modeling the nexus across water supply, power generation and environment systems using the system dynamics approach: Hehuang Region, China. J Hydrol, 543, 344-359, 2016.

Herman, J.D., Reed, Zeff, H.B., Characklis, G.W., 2015. How Should Robustness Be Defined for Water Systems Planning under Change? J. Water Resour. Plan. Manag. 141, 04015012.

Kasprzyk, J. R., Nataraj, S., Reed, P. M., and Lempert, R. J. (2013). "Many objective robust decisions making for complex environmental systems undergoing change."

Liu, W., Zhu, F., Chen, J., Wang, H., Xu, B., Song, P., ... & Li, J. (2019). Multi-objective optimization scheduling of wind–photovoltaic–hydropower systems considering riverine ecosystem. Energy conversion and management, 196, 32-43.

Pindyck, R.S., 2015. The Use and Misuse of Models for Climate Policy (Working Paper No. 21097). National Bureau of Economic Research.

---

## Short Comment (SC1) · 12 Aug 2020

Dear editor and authors,

This paper presents an interesting and significant contribution for coevolution process and feedback loop of water resources system considering the purpose of sustainable development by developing EEF nexus model, optimal approach, and system dynamic model. Sustainable development is always an important issue and will never outdated because it related to people's lives, and excessive social development undoubtedly

has a bad effect on environmental protections. The subject of the paper addresses an important scientific issue for both water resources and reservoir management, and well fits the scope of HESS. Overall, this paper is well-written, has a solid theoretical basis, clear purpose, and reasonable methods. I'm interested in this topic. This short comment may not represent the overall review but I expect my below comment can help strengthen the paper and further improve its quality.

Major concerns:

The methodology section includes three main parts: outline of EEF nexus, optimal modeling approach, and system dynamic model. The methodology is of great merit and technique, but it seems to be lacking in the connection between those approaches. In other words, how to integrate the optimal model with the outline of the EEF and system dynamic model? What is the mutual relationship between those methods? Perhaps the flowchart of the overall research framework can help. In my opinion, the overall methodology refers to the theory of complex adaptive system (CAS) as you mentioned in the Introduction section (Line 87-89) because EEF nexus is substantially a complex system and, the best status of both whole system and its agents must be attained and the optimal methods are required. The three agents in this study are just each module of EEF nexus. Considering this, the adaptive process of a complex system is substantially the optimal process and each agent has their own adaptive behaviors. The optimal model presented by authors with clear objectives, constraints and solutions, which does very well. So I suggest the authors link the outline of the EEF nexus and optimal model with the theory of complex adaptive system and demonstrate it in the flowchart.

Another major concern is the presentation of the system dynamic model. How to apply in the real case study is not clear. The driving factor of system coevolution is usually either interact within the agents or the effects of external conditions on both whole systems or agents. The authors addressed the "pendulum model" to illustrate the different situations of social development, and I guess that it is the latter driving factor

of system coevolution I stated before. It is also the driving factor of starting the adaptive adjustment process. But in the manuscript, I didn't find the relative statements, which to some extent is confused. This is the key point of the manuscript and authors should also add it in the flowchart. The introduction section should also contain such statements. The social changes drive the process of system coevolution and generate the feedback loop that is well presented in Fig.4. However, as far as I know, feedback regulations should contain cause-and-effect feedback loops that is reflected by either positive or negative. For example, increased social index causes increased water demand/supply, which further increases social index. This is a positive feedback loop. But some variables may be intensified. Negative feedback loops can eventually make the system more stable. This is the fundamental component of system dynamic model and authors should add such figure in the manuscript. Meanwhile, authors made a great contribution of feedback loop in the real case study. If the cause-and-effect feedback loop is in the manuscript, the statement about positive/negative feedback loop should be also included in the corresponding result section.

Minor concerns:

L16: What does the "scenario" refer to?

L30-31: The author stated that sustainable development is "ambiguity" but why does the author say "the new target"? It seems contradictory.

L129: by developing and controlling water...

L130: At once it is also indispensable.....: I guess "it" refers to water resources.

L165: resulting in a declining population: Please check if the statement is correct or add a reference. Row 8 Column 3 in Table2: Vegetation has something to do with crops?

L421-422: What do you mean by "reservoirs are of relatively high robustness for ecological module"?

L461: ULRB is both heavy industrial city, not Guilin city

P18: the last row in the first column: Initial ecological streamflow because initial stream-flow includes not only ecological issues but also for flood control, shipping, etc.

If the above concerns are considered, I believe that the quality of the paper will be greatly improved. I'm looking forward to the responses to my comments.

---

## Referee Comment (RC2) · Anonymous Referee #2 · 31 Aug 2020

This study seeks to provide insights into co-managing economic, ecological and food production objectives in a river basin using multi-criteria optimisation and systems dynamics modelling, with a case study in the Guijiang River Basin, China. I have a number of major concerns with this work:

(1) I don't see what is transdisciplinary about this study, i.e. where non-academic actors were involved in any part of the research. I don't think it fits with this special issue.

(2) The chain of models the authors employ is very complex and not always transpar-

ent; this obscures a lot of the uncertainty in those models.

I would have wished for more sensitivity analyses as those done for the "theta" parameter, which –as the authors rightly say – probably reflects most of the decision uncertainty as they set it up, but misses uncertainty in numerically representing the decision processes themselves in the first place, and uncertainty in the underlying hydrological models. More specifically:

L501-503: Not automatically, only when parameter uncertainty is low. When different parameter sets can lead to similarly "good" matches between simulations and observations – which I would expect for such a complex model – then anyone "optimal" parameter set cannot expected to reliably predict into the future – only if parameter uncertainty is propagated (not to mention other neglected uncertainties).

Sections 3.3.2-3.3.5: Here I'm missing an assessment of the numerical stability of the optimisation algorithm. For example, how unique is the solution found at each timestep, and how would small deviations change the dynamics presented here?

L689: "Can be mathematically expressed" – yes, but how do we know this expression is sensible?

(3) I find the goals ascribed to the 3 modules quite limited (e.g. illustrated in Figure 3). They should at least be underpinned by a thorough literature research that justifies why focussing on these is sensible. It should also be at least discussed what is left out and with what likely effect. For example in the ecological module, I'm missing a goal related to agricultural pollution and goals related to other ecosystem elements besides vegetation. And "minimum alteration of natural flow" and "maximum sewage treatment" are not so much ecosystem related goals but measures to achieve some other goals. On the goals of the food system another reviewer already commented eloquently, to which I would add the path-dependency build into the model when one of the goals is meat production and not some more flexible agricultural production responding to possibly changing diets. Some comments on the equations:

L145: In what ways is this growth rate "natural"?

Equation 2d: Are the constant considered universal? And what would be the justification for that?

Equation 3: How can the "natural flow" ever be determined?

Equation 7: How do the authors make sure that the data for variable W_L really isolate the water use of livestock? I can't imagine this is metered.

L302-304: Argument and justification for numerical algorithm unclear.

(4) The research gap that the authors wish to address doesn't become clear in the introduction. This section contains a lot of repetitions which makes it hard to understand the motivation for this study. It also remains unclear why the authors chose to frame their study in an economy-ecology-food nexus and not a more popular nexus variant reviewed in the introduction. Also:

L33-35: Singling out productive activities is downplaying the importance of water here.

L49: The concept of environmental stewardship should be introduced more carefully here.

L63: Why single out "optimisation" here?

(5) The language needs a thorough revision, e.g.:

L30: What's meant by "ambiguity and applicability" here?

L38-40: Sentence unclear.

L44-46: Argument unclear.

L76: I don't think these authors "achieved sustainable development"

L431-432: Meaning of "restoration" unclear.

(6) All in all, I'm left wondering what we have learned from this study that we didn't know

before, and hence hasn't been built in to the model in the first place. The conclusions are not informative in this respect.